# Development and validation of a multi-dimensional diagnosis-based comorbidity index that improves prediction of death in men with prostate cancer: Nationwide, population-based register study

**Marcus Westerberg**[1]*, **Sandra Irenaeus**[2,3], **Hans Garmo**[1], **Pär Stattin**[1], **Rolf Gedeborg**[1]

**1** Department of Surgical Sciences, Uppsala University, Uppsala, Sweden, **2** Department of Immunology, Genetics and Pathology, Uppsala University, Uppsala, Sweden, **3** Regional Cancer Center Midsweden, Uppsala, Sweden

* marcus.westerberg@uu.se

**Data Availability Statement:** Data used in the present study was extracted from the Prostate

## Abstract

Assessment of comorbidity is crucial for confounding adjustment and prediction of mortality in register-based studies, but the commonly used Charlson comorbidity index is not sufficiently predictive. We aimed to develop a multidimensional diagnosis-based comorbidity index (MDCI) that captures comorbidity better than the Charlson Comorbidity index. The index was developed based on 286,688 men free of prostate cancer randomly selected from the Swedish general population, and validated in 54,539 men without and 68,357 men with prostate cancer. All ICD-10 codes from inpatient and outpatient discharges during 10 years prior to the index date were used to define variables indicating frequency of code occurrence, recency, and total duration of related hospital admissions. Penalized Cox regression was used to predict 10-year all-cause mortality. The MDCI predicted risk of death better than the Charlson comorbidity index, with a c-index of 0.756 (95% confidence interval [CI] = 0.751, 0.762) vs 0.688 (95% CI = 0.683, 0.693) in the validation cohort of men without prostate cancer. Men in the lowest vs highest MDCI quartile had distinctively different survival in the validation cohort of men with prostate cancer, with an overall hazard ratio [HR] of 5.08 (95% CI = 4.90, 5.26). This was also consistent within strata of age and Charlson comorbidity index, e.g. HR = 5.90 (95% CI = 4.65, 7.50) in men younger than 60 years with CCI 0. These results indicate that comorbidity assessment in register-based studies can be improved by use of all ICD-10 codes and taking related frequency, recency, and duration of hospital admissions into account.

## Introduction

Accurate measures of comorbidity are needed to predict mortality, control confounding and/ or define relevant population strata in register-based observational studies. This is highly

Cancer Database Sweden (PCBaSe), which is based on the National Prostate Cancer Register (NPCR) of Sweden and linkage to several national health-data registers. The data cannot be shared publicly because the individual-level data contain potentially identifying and sensitive patient information and cannot be published due to legislation and ethical approval (https://etikprovningsmyndigheten.se). Use of the data from national health-data registers is further restricted by the Swedish Board of Health and Welfare (https://www.socialstyrelsen.se/en/) and Statistics Sweden (https://www.scb.se/en/) which are Government Agencies providing access to the linked healthcare registers. The data will be shared on reasonable request in an application made to any of the steering groups of NPCR and PCBaSe (contact npcr@npcr.se). For detailed information, please see www.npcr.se/in-english, where registration forms, manuals, and annual reports from NPCR are available alongside a full list of publications from PCBaSe. The code used for the analyses can be accessed via the following DOI (0.5281/zenodo.8210683; https://zenodo.org/doi/10.5281/zenodo.8210683).

**Funding:** PS received funding from Swedish Cancer Society (grant number 2022-2051, https://www.cancerfonden.se/) and Region Uppsala (www.regionuppsala.se/). The funders had no role in study design, data collection and analysis, decision to publish, or preparation of the manuscrip.

**Competing interests:** The authors have declared that no competing interests exist.

relevant for studies of men with prostate cancer, who often are above 65 years of age and have a high risk of death from other causes. The widely used Charlson comorbidity index and modifications thereof are based on the occurrence (Yes/No) of ICD-codes for a limited number of pre-specified diagnoses [1–3] and a large portion of men in prostate cancer studies have Charlson comorbidity index 0 [4–7], which hampers the prediction of mortality risk [5,8,9]. The restriction to a limited number of ICD-codes means that potentially relevant information from other codes that may be predictive is lost. A recently created drug comorbidity index avoided such selection by including all filled prescriptions up to a year before index date and clearly outperformed the Charlson comorbidity index [10]. Furthermore, there is more information in administrative health care registers than the mere occurrence of diagnostic codes. Other aspects such as frequency of occurrence of a code, time since latest appearance of a code (recency), and total duration of hospital admission associated with a code may also be predictive [5].

## Aim of the study

The aim of this study was to create a comorbidity index to be used in observational register-based studies. We hypothesized that predictive ability could be improved by extracting multiple quantitative aspects of all available diagnostic codes in a national patient register using a data-driven approach to identify the most important codes and quantitative aspects of code occurence. We compared the ability of this novel index to predict long-term mortality with the Charlson comorbidity index in men with and without prostate cancer.

## Materials

We used data from the Prostate Cancer data Base Sweden version 5 (PCBaSe 5) [11]. In PCBaSe 5, the National Prostate Cancer Register of Sweden has been linked by use of the unique Swedish Personal Identity Number to other healthcare registers held by the Swedish National Board of Health and Welfare, such as the Swedish Cancer Register, the Cause of Death Register [12], the National Prescribed Drug Register and the National Patient Register [13]. The capture rate of the National Prostate Cancer Register is above 96% compared to the Swedish Cancer register to which reporting is mandated by law [14]. The primary aim of NPCR is to provide data for quality assurance for cancer carer and adherence to National guidelines [15,16]. There are only modest differences in demographics, cancer treatment, comorbidity, and mortality between the men in NPCR and men only registered in the Cancer Register, indicating that information in NPCR can be generalized to all men with prostate cancer in Sweden.

PCBaSe 5 includes men diagnosed with prostate cancer between 1998 and 2020. For each prostate cancer case PCBaSe also includes five randomly selected comparison men, free from prostate cancer at the date of diagnosis of the corresponding case (index date), matched on birth year and county of residence.

Data was collected between January 1, 1998, and December 31, 2021, and accessed for research purposes June 1, 2022. The study population included all men in PCBaSe 5 with an index date between January 1, 2008, and December 31, 2014. The development cohort included all comparison men in PCBaSe 5 with an index date between January 1, 2008 and December 31, 2013. Two validation cohorts were used. The first cohort included all comparison men in PCBaSe 5 with an index date in 2014, and the second cohort all prostate cancer cases diagnosed between January 1, 2008, and December 31, 2014. Temporal rather than random splitting of the data is expected to reduce the risk for overly optimistic estimates of model performance.

All codes according to the 10th revision of International Statistical Classification of Diseases and Related Health Problems (ICD-10) registered as related to hospitalizations or specialist outpatient visits up to 10 years prior to the index date were extracted from the inpatient and specialist outpatient sub-registers of the National Patient Register. The National Patient Register comprises information on all in-hospital care and out-patient specialist care in Sweden. It has nation-wide coverage of in-patient care since 1987 and specialized outpatient care since 2001. During the study period, diagnoses were recorded according to the Swedish clinical modification of ICD-10 which has few modifications compared to the original ICD-10 version. Validation studies indicate that coding accuracy is diagnosis-specific [17–19].

The Swedish Prescribed Drug Register, used in this study to calculate a drug comorbidity index, contains details of all prescriptions dispensed in Sweden since July 1, 2005. Drugs are identified by a unique identifier for each specific combination of brand name, substance, formulation and package. The register only includes filled prescriptions, and not medicines sold over the counter or medicines administered directly to the patient during in-patient care, out-patient care or primary care.

Dates of death until 31 December 2020 were extracted from the Cause of Death Register for the follow-up of mortality. The Swedish cause of death register contains information on all deaths of Swedish residents since 1952.

## Methods

We developed a multidimensional diagnosis-based comorbidity index (MDCI) primarily to predict the risk of death of all causes within 10 years after each individuals index date, based on ICD-10 codes from inpatient stays and specialist outpatient visits. In complementary analyses the predictive performance with 1- and 5-year follow-up of mortality after the index date was also evaluated, using the same approach. The study adhered to the Transparent reporting of a multivariable prediction model for individual prognosis or diagnosis (TRIPOD) guidelines (S1 Appendix).

The Swedish Ethical Review Authority approved of the study [220–03437 and the need for consent was waiwed. Data was pseudonymized by the National Board of Health and Welfare prior to being delivered to the researchers.

### MDCI development strategy

The overall strategy was to derive predictors describing occurrence, frequency, recency, and duration of hospital admission of all ICD-10 codes identified in the National Patient Register, include all potential predictors in a Cox regression model, and use a variable selection strategy to minimise overfitting. To compute a patient's MDCI, one sums the coefficients from this model related to the selected predictors based on the ICD-10 codes observed for each patient during the 10 years preceding the index date. In an exploratory analysis using the development dataset we also evaluated the impact of length of lookback period for both the Charlson comorbidity index and the MDCI, alone and in combination with the drug comorbidity index. The process is described below and in greater detail in S2 Appendix.

### ICD-10 code granularity

The character positions in the ICD-10 code represent subcategories of a disease or condition. In practice, this allows registering of diagnoses with codes with fewer characters than the most detailed level specified by the ICD-10 coding system, or using a trailing "9" to indicate "unspecified". The use of such truncated codes may also vary between settings and over time. To address this the registered ICD-10 codes were first processed in a data management step

involving code truncation and elongation, pruning of unnecessary codes, and filtering codes present in at least 0.01% of the development cohort (**S1 Fig** and **S2 Appendix**).

## Definition of predictors

Each ICD-10 code could contribute with information on four levels depending on the number of characters used. For example, the code I731 (thromboangiitis obliterans) could contribute information based on two (I7), three (I73), four (I731), and five characters (I7319). For each of the four variants, code-specific predictors of mortality were generated and used in the regression models, describing occurrence, frequency, recency, and total duration of hospital admission in the 10 years preceding the index date. One predictor definition reflected simple occurrence of a code as primary diagnosis, and another indicated occurrence as either primary or secondary diagnosis. The predictors for frequency, recency, and duration were only based on codes registered as primary diagnosis. Frequency was categorized as occurrence on $\geq 2$, $\geq 3$, or $\geq 4$ unique dates. Recency was categorized as occurence within 90, 180 and 365 days prior to the index date. Duration of hospital admission was categorized by dummy variables indicating if the total number of days in hospital exceeded 7 or 14 days. These cutoffs were prespecified based on clinical reasoning with the intention to separate patients with short observational and mainly diagnostic stays from patients with severe conditions and/or reduced functional status.

## Variable selection and model fitting

We used a Cox proportional hazards regression model with all the predictors as covariates [8,20] and applied regularization by use of the elastic net, [21,22], to simultaneously perform both variable selection and handle collinearity. Ten-fold cross-validation over a grid of 100 values of the hyper-parameter that controls the amount of penalization was used to identify the model with the highest concordance index (c-index) [23,24]. The analysis was performed using R version 4.1.3 and *glmnet* version 4.1–2 [25].

## Validation

The performance of the MDCI was compared to the Charlson comorbidity index and the drug comorbidity index in the two validation cohorts separately, using the c-index and calibration curves [26]. We computed the c-index for mortality at 1, 5 and 10 years of follow-up after the index date, and computed 95% confidence intervals (CIs) for the c-indices using a bootstrap procedure with 1000 replications. The Charlson comorbidity index was calculated based on ICD-10 codes registered as a primary or secondary diagnosis in the National Patient Register during the 10-year period preceding the index date [27]. When the Charlson comorbidity index was calculated in the validation cohort with prostate cancer cases we excluded ICD-10 codes for prostate cancer (C61), and metastases (C77-80) if they were registered in conjunction with C61. This is in line with previous adaptations of the Charlson comorbidity index to cancer populations [28]. The drug comorbidity index was calculated based on all drug prescriptions registered in the National Prescribed Drug Register during the 365-day period preceding the index date [8].

## Survival analysis

Survival was assessed with Kaplan-Meier curves and 1-year hazard ratios (HR) estimated using a Cox proportional hazards model [29]. Analyses were stratified by age, Charlson comorbidity index, the drug comorbidity index, and the MDCI. The MDCI was categorized by using the

25%, 50% and 75% quantiles of the distribution of the MDCI in the development cohort. The same procedure was applied to categorize the drug comorbidity index.

## Results

### Study participants

The study population consisted of 409,584 men with an age range of 32–102 years (**Table 1**). The development and validation cohorts were similar in terms of age, Charlson comorbidity index, drug comorbidity index, and ICD-10 code chapters registered during the 10-year period preceding the index date. Median length of follow-up for mortality was 8 years in the development cohort and 30% (85,468 men) died within 10 years (**Table 2**). There were 10,325 unique ICD-10 codes registered. Of these, 45% (5612) were observed in at least 0.01% of the development cohort and these codes generated 56,120 potential predictors of mortality (**S1 Fig**).

### Selected codes and code dimensions

The MDCI derived after the variable selection process included information from 978 unique ICD-10 codes, out of which 58% (564) had 4 or 5 characters. These codes generated 1543 corresponding predictors of mortality, out of which 56% (870) reflected occurrence, 17% (261) frequency, 17% (264) recency, and 10% (148) duration of hospital admission (**S2 Fig**).

As examples, the predictors for mortality derived from ICD-10 codes for cardiovascular diseases are demonstrated (**Fig 1**). The example illustrates how predictors reflecting occurrence, frequency, recency, and duration of hospital admission of ICD-10 codes independently contributed with predictive information to a varying degree depending on the specific diagnosis. For cardiovascular diseases additional predictive ability was sometimes also added from codes with 5 characters. For diseases such as arrythmia, cardiomyopathy, cardiac arrest, and heart failure, additional information was often obtained using 5 characters and from all four dimensions of predictors. Similar distributions of predictors related to all other diseases are shown in (**S3 Fig**). Some categories that were not expected to contribute, such as carcinoma in situ (D0) for malignancies, in fact added predictive information.

In complementary analyses we derived the MDCI also using 1 and 5 years of follow-up for mortality after the index date. The number of informative diagnosis codes decreased with shorter follow-up. For the 1-year MDCI, 369 unique ICD-10 codes were selected, compared to 885 unique ICD-10 codes for the 5-year MDCI (**S2 Fig**). Based on these ICD-10 codes there were 547 predictors selected for the 1-year MDCI and 1378 for the 5-year MDCI. The distributions between factors reflecting occurrence, frequency, recency, and duration were comparable to that seen for the MDCI developed from 10-year follow-up for mortality (**S2 and S3 Figs**).

The c-indices from the cross-validation procedure can be found in **S1 Table**. All included predictors and their coefficients are listed in **S2 Table**.

### Model discrimination and calibration

The MDCI improved discrimination both in men without (c-index = 0.756; 95% CI = 0.751, 0.762) and with prostate cancer (c-index = 0.702; 95% CI = 0.699, 0.706) compared to the Charlson comorbidity index (c-index = 0.688; 95% CI = 0.683, 0.693) and (c-index = 0.628; 95% CI = 0.625, 0.631), respectively (**Table 2**). An improvement was also seen when compared to the drug comorbidity index (c-index = 0.732; 95% CI = 0.727, 0.738) and (c-index = 0.666; 95% CI = 0.662, 0.670), respectively. The discrimination of the MDCI was slightly higher when follow-up for mortality was short. The c-index at 1-year of follow-up was 0.841 (95%

**Table 1. Baseline characteristics of the development cohort and the two validation cohorts in Prostate Cancer data Base Sweden (PCBaSe) 5.**

| | All men | | Development cohort | | Validation cohort without prostate cancer | | Validation cohort with prostate cancer | |
|---|---|---|---|---|---|---|---|---|
| | (N = 409,584) | | (N = 286,688) | | (N = 54,539) | | (N = 68,357) | |
| **Age at index date** (years), n (%) | | | | | | | | |
| <60 | 39,175 | (10) | 27,484 | (10) | 5,160 | (9) | 6,531 | (10) |
| 60–69 | 153,867 | (38) | 107,577 | (38) | 20,640 | (38) | 25,650 | (38) |
| 70–79 | 144,161 | (35) | 99,427 | (35) | 20,698 | (38) | 24,036 | (35) |
| ≥80 | 72,381 | (18) | 52,200 | (18) | 8,041 | (15) | 12,140 | (18) |
| **Index year**, n (%) | | | | | | | | |
| 2008–2010 | 174,958 | (43) | 145,789 | (51) | 0 | (0) | 29,169 | (43) |
| 2011–2013 | 169,139 | (41) | 140,899 | (49) | 0 | (0) | 28,240 | (41) |
| 2014 | 65,487 | (16) | 0 | (0) | 54,539 | (100) | 10,948 | (16) |
| **Charlson comorbidity index (CCI)**, n (%) | | | | | | | | |
| 0 | 263,067 | (64) | 183,619 | (64) | 34,587 | (63) | 44,861 | (66) |
| 1 | 59,668 | (15) | 42,268 | (15) | 7,877 | (14) | 9,523 | (14) |
| 2 | 49,125 | (12) | 33,997 | (12) | 6,699 | (12) | 8,429 | (12) |
| 3+ | 37,724 | (9) | 26,804 | (9) | 5,376 | (10) | 5,544 | (8) |
| **Drug comorbidity index (DCI)**, n (%) | | | | | | | | |
| <Quartile 1 | 108,829 | (27) | 79,617 | (28) | 15,271 | (28) | 13,941 | (20) |
| Quartile 1—Quartile 2 | 93,250 | (23) | 63,727 | (22) | 12,341 | (23) | 17,182 | (25) |
| Quartile 2—Quartile 3 | 105,055 | (26) | 71,672 | (25) | 13,585 | (25) | 19,798 | (29) |
| >Quartile 3 | 102,450 | (25) | 71,672 | (25) | 13,342 | (24) | 17,436 | (26) |
| **Occurrence of ICD-10 code chapter**[a], n (%) | | | | | | | | |
| A or B (infectious and parasitic diseases) | 102,450 | (25) | 71,672 | (25) | 13,342 | (24) | 17,436 | (26) |
| C or D (neoplasms and diseases of the blood and blood-forming organs) | 43,241 | (11) | 29,754 | (10) | 6,463 | (12) | 7,024 | (10) |
| E (endocrine, nutritional, and metabolic diseases) | 94,788 | (23) | 61,857 | (22) | 13,144 | (24) | 19,787 | (29) |
| F (mental and behavioral disorders) | 80,900 | (20) | 56,531 | (20) | 12,069 | (22) | 12,300 | (18) |
| G (diseases of the nervous system) | 38,872 | (9) | 27,418 | (10) | 6,079 | (11) | 5,375 | (8) |
| H (ear, nose, or throat diseases) | 60,071 | (15) | 41,604 | (15) | 8,930 | (16) | 9,537 | (14) |
| I (diseases of the circulatory system) | 149,747 | (37) | 10,2475 | (36) | 21,647 | (40) | 25,625 | (37) |
| J (diseases of the respiratory system) | 64,760 | (16) | 44,829 | (16) | 9,344 | (17) | 10,587 | (15) |
| K (diseases of the digestive system) | 113,582 | (28) | 78,268 | (27) | 15,449 | (28) | 19,865 | (29) |
| L (diseases of the skin and subcutaneous tissue) | 70,995 | (17) | 47,666 | (17) | 11,009 | (20) | 12,320 | (18) |
| M (diseases of the musculoskeletal system and connective tissue) | 132,057 | (32) | 89,010 | (31) | 19,795 | (36) | 23,252 | (34) |
| N (diseases of the genitourinary system) | 98,379 | (24) | 63,403 | (22) | 12,432 | (23) | 22,544 | (33) |
| R (symptoms, signs, and abnormal clinical and laboratory findings, not elsewhere classified) | 182,464 | (45) | 118,543 | (41) | 24,744 | (45) | 39,177 | (57) |
| S or T (injury, poisoning and certain other consequences of external causes) | 125,534 | (31) | 86,211 | (30) | 18,248 | (33) | 21,075 | (31) |
| Z (factors influencing health status and contact with health services | 207,660 | (51) | 141,946 | (50) | 29,645 | (54) | 36,069 | (53) |
| Others[b] | 4419 | (1) | 2842 | (1) | 868 | (2) | 709 | (1) |

[a]Based on cleaned codes prior to the processing and filtering of codes present in at least 0.01% of men in the development cohort.

[b]Others include codes starting with letters O, P, Q, U, and Y observed in N = 4, 95, 3,506, 827, and 6 subjects, respectively.

CI = 0.830, 0.852) in men without prostate cancer and 0.793 (95% CI = 0.786, 0.801) in men with prostate cancer (**Table 2**).

The MDCI based on 1-year of follow-up for mortality correlated well with the MDCI based on a 10-year follow-up (correlation coefficient 0.90–0.93) (**S4 Table**). The discrimination was

**Table 2. Duration of follow-up for mortality after the index date, number of deaths, and model discrimination in development and validation cohorts.**

| | All | | Development cohort | | Validation cohort without prostate cancer | | Validation cohort with prostate cancer | |
|---|---|---|---|---|---|---|---|---|
| | **(N = 409,584)** | | **(N = 286,688)** | | **(N = 54,539)** | | **(N = 68,357)** | |
| **Follow-up** (years), median (IQR) | 8 | (6–10) | 9 | (7–11) | 6 | (6–7) | 8 | (6–10) |
| ≤ 1 year, n (%) | 13,000 | (3) | 8,643 | (3) | 1,439 | (3) | 2,918 | (4) |
| 1–5 years, n (%) | 53,741 | (13) | 36,215 | (13) | 6,200 | (11) | 11,326 | (17) |
| 5–10 years, n (%) | 227,590 | (56) | 144 288 | (50) | 46,900 | (86) | 36,402 | (53) |
| >10 years, n (%) | 115,253 | (28) | 97,542 | (34) | 0 | (0) | 17,711 | (26) |
| **Number of deaths** | | | | | | | | |
| At 1 year | 13,000 | (3) | 8,643 | (3) | 1,439 | (3) | 2,918 | (4) |
| At 5 years | 66,741 | (16) | 44,858 | (16) | 7,639 | (14) | 14,244 | (21) |
| At 10 years | 118,601 | (29) | 85,468 | (30) | 10,115 | (19) | 23,018 | (34) |
| **Model discrimination** | | | C-index (95% CI) | | C-index (95% CI) | | C-index (95% CI) | |
| *Multi-dimensional diagnosis-based comorbidity index (MDCI) developed for 1-year mortality evaluated using* | | | | | | | | |
| 1 year of follow-up after index date | | | 0.823 (0.818–0.828) | | 0.829 (0.815–0.840) | | 0.782 (0.773–0.791) | |
| 5 years of follow-up after index date | | | 0.741 (0.738–0.743) | | 0.747 (0.740–0.753) | | 0.702 (0.697–0.706) | |
| 10 years of follow-up after index date | | | 0.699 (0.697–0.701) | | 0.728* (0.723–0.734) | | 0.679 (0.675–0.682) | |
| *MDCI developed for 5-year mortality evaluated using* | | | | | | | | |
| 1 year of follow-up after index date | | | 0.832 (0.827–0.837) | | 0.842 (0.83–0.852) | | 0.794 (0.786–0.802) | |
| 5 years of follow-up after index date | | | 0.766 (0.763–0.768) | | 0.767 (0.761–0.773) | | 0.719 (0.715–0.723) | |
| 10 years of follow-up after index date | | | 0.725 (0.723–0.726) | | 0.750* (0.744–0.755) | | 0.696 (0.693–0.700) | |
| *MDCI developed for 10-year mortality evaluated using* | | | | | | | | |
| 1 year of follow-up after index date | | | 0.832 (0.827–0.837) | | 0.841 (0.83–0.852) | | 0.793 (0.786–0.801) | |
| 5 years of follow-up after index date | | | 0.769 (0.766–0.771) | | 0.772 (0.766–0.778) | | 0.723 (0.719–0.727) | |
| 10 years of follow-up after index date | | | 0.734 (0.732–0.735) | | 0.756* (0.751–0.762) | | 0.702 (0.699–0.706) | |
| *Charlson comorbidity index (CCI) evaluated using* | | | | | | | | |
| 1 year of follow-up after index date | | | 0.752 (0.747–0.757) | | 0.758 (0.746–0.770) | | 0.683 (0.673–0.693) | |
| 5 years of follow-up after index date | | | 0.695 (0.693–0.698) | | 0.701 (0.695–0.707) | | 0.637 (0.633–0.641) | |
| 10 years of follow-up after index date | | | 0.668 (0.666–0.670) | | 0.688* (0.683–0.693) | | 0.628 (0.625–0.631) | |
| *Drug comorbidity index (DCI) evaluated using* | | | | | | | | |
| 1 year of follow-up after index date | | | 0.799 (0.794–0.804) | | 0.804 (0.792–0.816) | | 0.731 (0.722–0.74) | |
| 5 years of follow-up after index date | | | 0.739 (0.736–0.741) | | 0.743 (0.737–0.749) | | 0.677 (0.672–0.682) | |
| 10 years of follow-up after index date | | | 0.701 (0.708–0.712) | | 0.732* (0.727–0.738) | | 0.666 (0.662–0.670) | |

*Follow-up restricted to a maximum of 7 years.

overall slightly lower, but still consistently higher compared to the Charlson comorbidity index and the drug comorbidity index (**Table 2**). The pattern was also consistent with shorter follow-up resulting in higher discrimination (**Table 2**).

Calibration curves comparing predicted and observed 10-year probabilities of death indicated good calibration of the MDCI, and also illustrates the increased separation of mortality risk provided by the MDCI compared to the Charlson comorbidity index (**Fig 2**). Calibration was similar for predicted 5-year mortality risk but less optimal for prediction of 1-year mortality, where the MDCI tended to underestimate mortality risk in strata with low mortality and overestimate the risk in strata with higher mortality risk (**S4 Fig**).

In complementary analyses, decreasing the length of the lookback period to 5 years or 1 year for the Charlson comorbidity index or for the MDCI clearly decreased the predictive ability, e.g. using a lookback of one year the C-index for the Charlson comorbidity index and

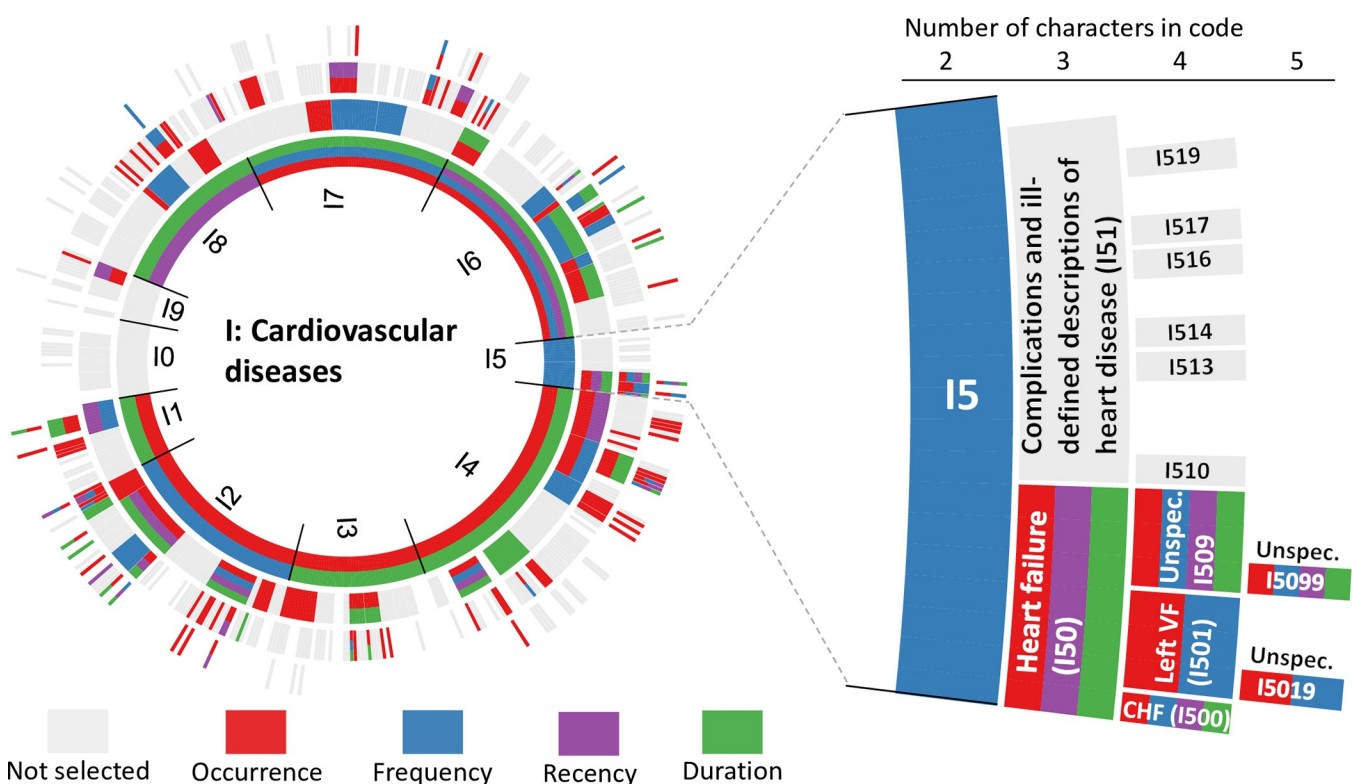

**Fig 1. Example of predictors selected in the prediction model.** ICD-10 codes for cardiovascular diseases are shown in the figure, specifically focusing on the ICD-10 code category I5 (heart failure and some heart disorders and diseases). The innermost circle represents ICD-10 codes with two characters and the outmost circle codes with five characters and grouped predictors. Each predictor variable (occurrence, frequency, recency, duration) derived for each ICD-10 code corresponds to a color-coded circle segment. A segment is colored if any coefficient within that group of predictors had a non-zero coefficient for that code, and grey if non-informative. I50 = heart failure, I51 = complications and ill-defined descriptions of heart disease, I500 = right ventricular failure, I501 = left ventricular failure, I509 = heart failure, unspecified, I510 = cardiac septal defect, acquired, I513 = intracardiac thrombosis, not elsewhere classified, I514 = myocarditis, unspecified, I516 = cardiovascular disease, unspecified, I517 = cardiomegaly, I519 = heart disease, unspecified, I5019 = left ventricular failure, unspecified, I5099 = heart failure, unspecified.

MDCI were 0.599 (95% CI: 0.597–0.601) and 0.637 (0.635–0.639), respectively (**S3 Table**). When the comorbidity indices were combined with the drug comorbidity index the overall predictive ability increased and the impact of lookback was less pronounced, although a lookback of 10 years produced the largest C-indices both for Charlson comorbidity index and the MDCI, 0.724 (95% CI: 0.722–0.725) and 0.751 (95% CI: 0.749–0.752), respectively (**S3 Table**).

## Added value for prediction of mortality in comparison with other comorbidity indices

The MDCI identified groups of comparison men with distinctively different survival probability in the validation cohort, also within strata according to the Charlson comorbidity index or the drug comorbidity index (**Fig 3**). This observation was consistent throughout all strata of age and Charlson comorbidity index (**S5 Fig**) and in men with prostate cancer (**S6 Fig**). A similar pattern was seen within strata of age and the drug comorbidity index (**S7 and S8 Figs**). The correlation between the MDCI and the drug comorbidity index was low (correlation coefficient ≤0.64) (**S3 Table**).

The MDCI was particularly good at discriminating risk of death at the extremes of the scale, i.e., for men in the top and bottom quartile of the MDCI. In the validation cohort consisting of men without prostate cancer, there was an increased risk of death in the highest vs

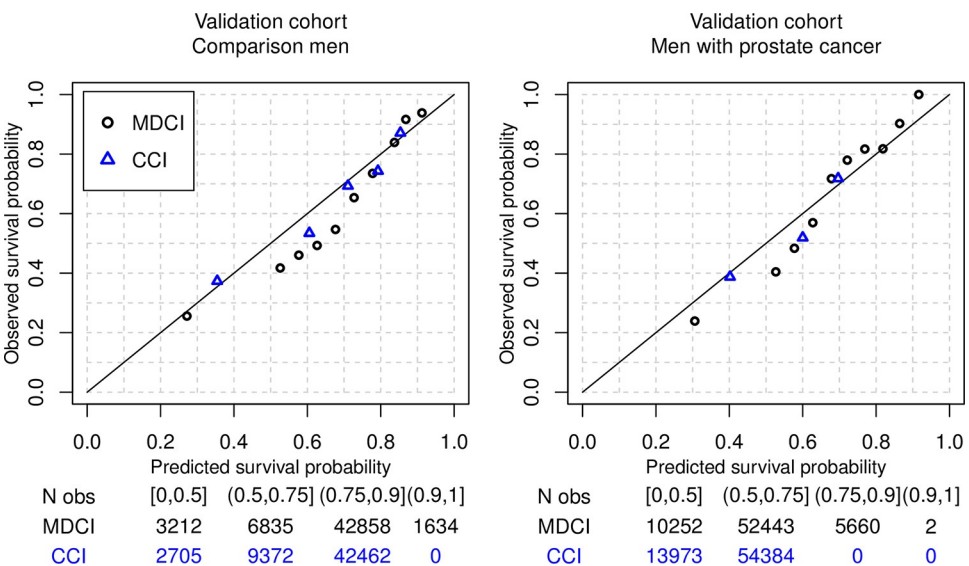

| N obs | [0,0.5] | (0.5,0.75] | (0.75,0.9] | (0.9,1] | | N obs | [0,0.5] | (0.5,0.75] | (0.75,0.9] | (0.9,1] |
|---|---|---|---|---|---|---|---|---|---|---|
| MDCI | 3212 | 6835 | 42858 | 1634 | | MDCI | 10252 | 52443 | 5660 | 2 |
| CCI | 2705 | 9372 | 42462 | 0 | | CCI | 13973 | 54384 | 0 | 0 |

**Fig 2. Calibration plots.** Observed 10-year mortality risk compared to predicted risk based on the MDCI or the Charlson comorbidity index among 54,539 men without prostate cancer and 68,357 men with prostate cancer. In each cohort, the calibration plot was obtained by computing the predicted 10-year survival probability for each individual using a Cox proportional hazards model including the MDCI or Charlson comorbidity index, respectively as predictor, and a corresponding estimate of the baseline hazard function. The observed survival probability was computed as the average 10-year survival estimated by the Kaplan-Meier curve within the intervals of the predicted survival probabilities using the cutoffs 0%, 50%, 55%, 60%, 65%, 70%, 75%, 80%, 85%, 90% and 100%. The predicted survival probability was similarly computed as the average of all individual survival probabilities among individuals within each interval. Probability intervals containing zero individuals are not shown.

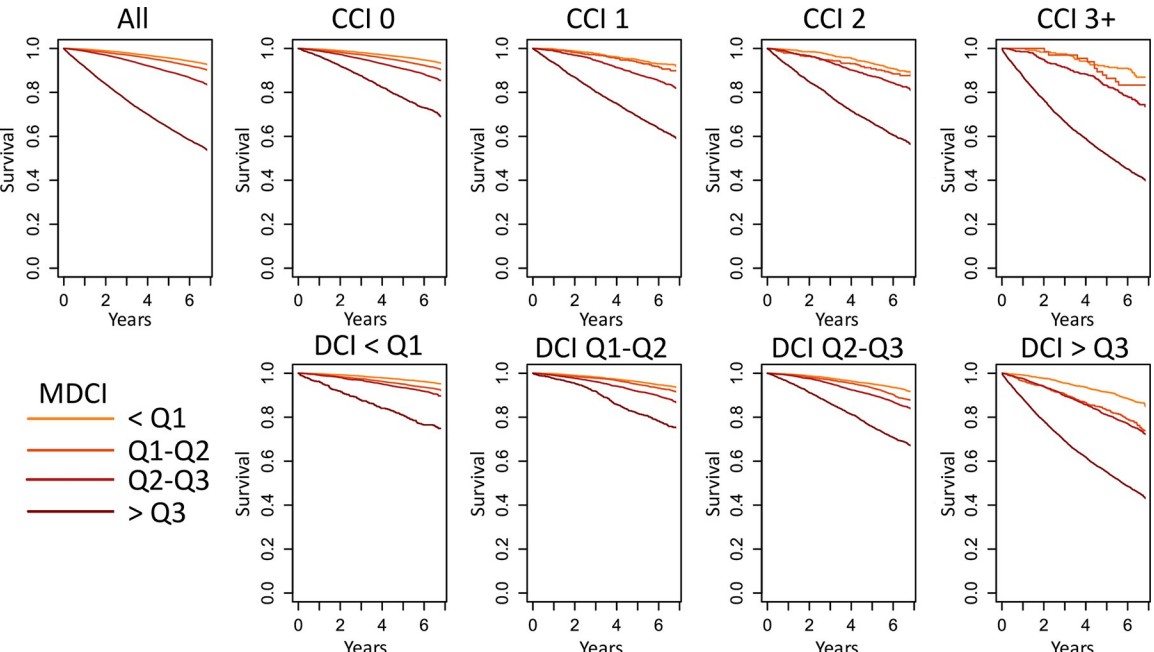

**Fig 3. Survival of men in the validation cohort consisting of men without prostate cancer.** The validation cohort has been split in subgroups based on the Charlson comorbidity index (0, 1, 2, 3+) and the drug comorbidity index (quartiles of the DCI). The survival within each subgroup is then shown stratified in quartiles of the MDCI developed using 10 years of follow-up for mortality.

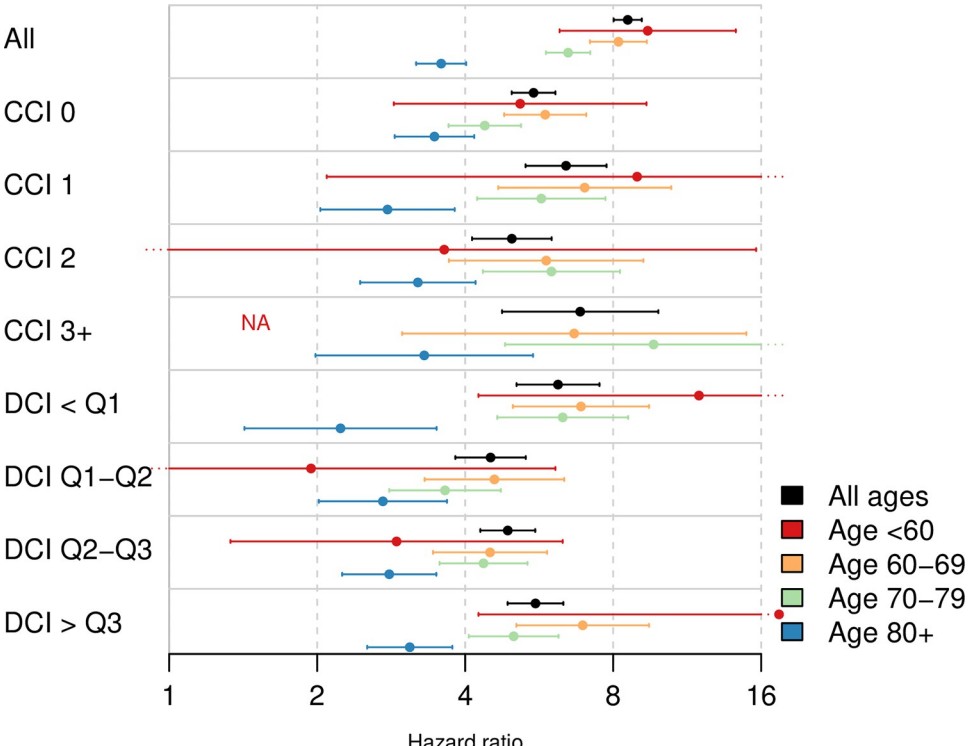

**Fig 4. 1-year hazard ratios for men in the validation cohort consisting of men without prostate cancer.** Stratified by age, Charlson comorbidity index (CCI) and drug comorbidity index (DCI) for the multi-dimensional diagnosis-based comorbidity index (MDCI) developed using 10 years of follow-up for mortality. Not available (NA) indicates that the hazard ratio could not be estimated due to small sample size and/or few events in each stratum.

lowest quartile of MDCI (HR = 8.56; 95% CI = 8.02, 9.15) (**Fig 4**). The difference was largest in the youngest age group and among those with low comorbidity based on the other scores. For men <60 years and Charlson comorbidity index = 0, there was a 5-fold difference in mortality (HR = 5.17; 95% CI = 2.86, 9.35). This difference was smaller in men > 80 years (HR = 3.51; 95% CI = 2.91, 4.23). The pattern of separation was consistent across strata of both age and the Charlson index and the drug comorbidity index (**Fig 4**). In men with prostate cancer the increase in risk of death according to MDCI was somewhat weaker in all men (HR = 5.08; 95% CI = 4.9, 5.26), men with CCI = 0 and <60 years of age (HR = 5.9; 95% CI = 4.65, 7.5), and CCI = 0 and age above 80 (HR = 2.09; 95% CI = 1.88, 2.32) (**S9 Fig**).

The MDCI using 1 and 5 years of follow-up for mortality resulted in comparable HRs in the two validation cohorts when comparing the quarters of the cohort with highest and lowest MDCI, and stratified men equally well (**S9 Fig**). For example, the MDCI developed using 1 year of follow-up for mortality clearly separated risk of death both for men without prostate cancer (HR = 5.71; 95% CI = 5.38, 6.05) and with prostate cancer (HR = 3.65; 95% CI = 3.52, 3.77). The MDCI using 5 years of follow-up indicated similar separation (HR = 7.64; 95% CI = 7.18, 8.13) and (HR = 4.7; 95% CI = 4.54, 4.86), respectively.

## Discussion

### Summary of main results

Our new multidimensional diagnosis-based comorbidity index (MDCI) based on occurrence, recency, frequency, and duration of hospital admissions for all ICD-10 codes predicted risk of

death better than the commonly used Charlson comorbidity index. This was demonstrated consistently across strata of both age and other comorbidity indices. Notably, the MDCI was also able to stratify men with Charlson comorbidity index = 0 into strata with clearly different risk of death. This supports the hypothesis that relevant quantitative information related to disease severity can be extracted from diagnosis codes in longitudinal register data. The prediction model developed using 10 years of follow-up performed better than models based on shorter follow-up.

## Comparison with other comorbidity indices

The commonly used versions of the Charlson comorbidity index [27,30] are based on the occurrence of a limited number of pre-specified diagnostic codes in an administrative health care register. In contrast, we assessed the predictive value of all ICD-10 codes available in the Swedish National Patient Registry without considering prior knowledge of the potential predictive value of the codes. In our analysis, we also included information on the number of occurrences during a specified time interval, the duration of hospitalization with a code as a main diagnosis, and the time interval from the latest hospital discharge to index date, as well as if the code was registered as a main or secondary diagnosis.

The concept of such a multi-dimensional comorbidity index has recently been demonstrated to be valid for intensive care based on 36 predefined comorbidity categories and predictors indicating the number of hospital admissions during the past 5 years, duration of hospitalization and time since latest admission [5]. This comorbidity index identified groups with clear differences in survival also within strata of the Charlson comorbidity index and age. Our study provides further support for the concept of a multi-dimensional diagnosis-based comorbidity index. Other attempts in this direction have been made. A recent study using Medicare claims linked with Health and Retirement Study data developed seven dichotomous markers of disease severity, including markers based on the number of disease-associated outpatient visits, emergency department visits, and hospitalizations made by an individual over a defined interval [31]. These markers did not, however, meaningfully predict outcomes such as decline in activities of daily living or mortality.

The strata of individuals with Charlson comorbidity index 0 is often large [4–6] but heterogeneous in terms of risk of death [5,8], which we confirmed in our study. The improved ability of the MDCI to discriminate risk of death in this group, and in subgroups defined by age, suggests that the MDCI should be preferred over the Charlson index when adjusting for comorbidity or predicting life expectancy in register-based studies of prostate cancer. Since diagnosis codes reflect information from hospitalizations or specialist out-patient care, even better prediction can possibly be achieved when combining the MDCI with the newly created drug comorbidity index [8] that summarizes information also from primary care. Our results suggest that the MDCI and the drug comorbidity index reflect different aspects of a patient's medical history and therefore should be used in combination, preferably with a lookback of 5–10 years for the MDCI.

## Strengths and limitations

Strengths of our study include that the development cohort comprised data from more than 280,000 men with a complete 10-year history of ICD-10 codes registered in a National Patient Register that has high validity [13]. Furthermore, we performed validation in two large and independent cohorts, one comprising men without prostate cancer alive at a later calendar period and the other comprising men with a prostate cancer diagnosis. Developing the MDCI in men without prostate cancer allowed for estimation of risk of death from all causes in

absence of prostate cancer diagnosis without introducing potential bias due to misclassification of the cause of death [32].

There are some limitations to our study. Linkages of rich sources at an individual level can be done in Sweden but not everywhere else so this is a limitation for generalization of this method to other countries and settings. The prediction models were developed in men selected as comparison men to men with prostate cancer, meaning that e.g. extrapolation of model parameters to women or specific other patient groups should be cautious. Associations between ICD-codes and mortality may also be context-sensitive because disease patterns, coding practices, and treatment strategies likely vary between countries and over time [10]. External validation is therefore needed in other health care settings. The principle for extraction of comprehensive information from the ICD-10 codes may therefore be more generalizable than the generated actual parameter estimates. The selection of ICD-10 codes and associated parameter estimates should be used cautiously in other settings and may also need to be revisited over time. If the purpose is to control selection bias and/or confounding in a study, it is likely as effective, or even more effective, to include specific ICD-10 codes and associated predictors and derive their weights internally for the specific study. This, however, requires a large sample size. The MDCI was compared to a recently published version of the Charlson index, specifically adapted to the Swedish Patient Register [27]. Comparisons to other comorbidity measures, including different adaptations of the Charlson comorbidity index and the Elixhauser comorbidity index, may yield different results.

The MDCI was developed and validated for use in register-based non-interventional studies and not for application on individual patients in clinical practice. While the study population was large, the results may be influenced by the restriction to men and the distribution of comorbidities in this particular population. The study population was therefore not well suited to elucidate the relative contribution of predictors based on occurrence, frequency, duration, and recency to the overall performance of the MDCI. The main purpose of this study was to provide proof of principle for a multidimensional diagnosis-based comorbidity index. Future research should assess the discrimination and calibration of the MDCI in a general population, including both men and women. In such a general population sample, it is of interest to try to identify what specific features of this new index that drives the improved predictive ability, so that the principle also can be adapted to other settings with less comprehensive data sources.

## Conclusions

A multidimensional diagnosis-based comorbidity index based on all ICD-10 diagnostic codes in a National Patient Register, including information of occurrence, recency, frequency, and duration of hospital admission of each code, clearly outperformed the commonly used Charlson comorbidity index in prediction of death in men with and without prostate cancer.

## Supporting information

**S1 Appendix. TRIPOD checklist.**
(DOCX)

**S2 Appendix. Supplementary methods.**
(DOCX)

**S1 Table. C-indices obtained from repeated cross-validation in the development cohort.**
(XLSX)

**S2 Table. Predictors and estimates in the multi-dimensional diagnosis-based comorbidity indices (MDCI) developed using 10 years of follow-up for mortality.**
(XLSX)

**S3 Table. Exploratory analysis assessing the impact on the C-index of the length of look-back for the Charlson comorbidity index and the multi-dimensional diagnosis-based comorbidity index (MDCI), alone or in combination with the drug comorbidity index.**
(XLSX)

**S4 Table. Correlation between the multi-dimensional diagnosis-based comorbidity indices (MDCIs) and between the MDCIs and the drug comorbidity index.**
(XLSX)

**S1 Fig. Overview of the code extraction and creation of predictors.**
(PDF)

**S2 Fig. The number and type of predictors selected by each of the multi-dimensional diagnosis-based comorbidity indices (MDCI) developed using 1, 5, and 10 years of follow-up for mortality.**
(PDF)

**S3 Fig. Selected ICD-10 codes.** ICD-10 codes with 2–5 characters in each circle (inner circle = 2 characters, outer circle = 5 characters) and grouped predictors (occurrence, frequency, recency, duration). Each predictor for each code corresponds to a circle segment and this segment is colored if any coefficient within that group of predictors was included in the multi-dimensional diagnosis-based comorbidity index developed using 1, 5, and 10 years of follow-up for mortality, respectively.
(PDF)

**S4 Fig. Calibration plots.** Observed 1, 5, and 10-year mortality risk compared to predicted risk based on the MDCI developed using 1, 5, and 10 years of follow-up for mortality, respectively, or the CCI.
(PDF)

**S5 Fig. Survival by the multi-dimensional diagnosis-based comorbidity indices (MDCI).** Developed using 10 years of follow-up for mortality vs the Charlson comorbidity index (CCI) in the validation cohort of comparison men.
(PDF)

**S6 Fig. Survival by the multi-dimensional diagnosis-based comorbidity indices (MDCI).** Developed using 10 years of follow-up for mortality vs the Charlson comorbidity index (CCI) in the validation cohort of men with prostate cancer.
(PDF)

**S7 Fig. Survival by the multi-dimensional diagnosis-based comorbidity indices (MDCI).** Developed using 10 years of follow-up for mortality vs the drug comorbidity index (DCI) in the validation cohort of comparison men.
(PDF)

**S8 Fig. Survival by the multi-dimensional diagnosis-based comorbidity indices (MDCI).** Developed using 10 years of follow-up for mortality vs the drug comorbidity index (DCI) in the validation cohort of men with prostate cancer.
(PDF)

**S9 Fig. 1-year hazard ratios for men in the validation cohorts of comparison men and men with prostate cancer.** Comparing men with a multi-dimensional diagnosis-based comorbidity index (MDCI) below Q1 vs men with MDCI above Q3. The analysis was also stratified by age, Charlson comorbidity index (CCI) and drug comorbidity index (DCI). NA indicates that the hazard ratio could not be estimated due to small sample size and/or few events in each stratum.
(PDF)

## Acknowledgments

This project was made possible by the continuous work of the National Prostate Cancer Register (NPCR) of Sweden steering group:

Ingela Franck Lissbrant, David Robinson, Johan Styrke, Johan Stranne, Jon Kindblom, Camilla Thellenberg, Andreas Josefsson, Ingrida Verbiené, Hampus Nugin, Stefan Carlsson, Anna Kristiansen, Mats Andén, Thomas Jiborn, Olof Ståhl, Olof Akre, Per Fransson, Eva Johansson, Magnus Törnblom, Fredrik Jäderling, Marie Hjälm Eriksson, Lotta Renström, Jonas Hugosson, Ola Bratt, Maria Nyberg, Fredrik Sandin, Fredrik Sandin, Maria Brus, Mats Lambe, Anna Hedström, Nina Hageman, Christofer Lagerros, Hans Joelsson, and Gert Malmberg.

## Disclaimer

R.G. is also employed by the Medical Products Agency (MPA). The MPA is a Swedish Government Agency. The views expressed in this article may not represent the views of the MPA.

## Author Contributions

**Conceptualization:** Marcus Westerberg, Sandra Irenaeus, Hans Garmo, Rolf Gedeborg.

**Data curation:** Hans Garmo, Pär Stattin.

**Formal analysis:** Marcus Westerberg.

**Funding acquisition:** Pär Stattin.

**Investigation:** Marcus Westerberg, Pär Stattin, Rolf Gedeborg.

**Methodology:** Marcus Westerberg, Hans Garmo, Rolf Gedeborg.

**Project administration:** Pär Stattin.

**Resources:** Pär Stattin.

**Software:** Marcus Westerberg.

**Validation:** Marcus Westerberg.

**Visualization:** Marcus Westerberg, Hans Garmo.

**Writing – original draft:** Marcus Westerberg, Rolf Gedeborg.

**Writing – review & editing:** Marcus Westerberg, Sandra Irenaeus, Hans Garmo, Pär Stattin, Rolf Gedeborg.

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
