## [Decision Letter · Decision Letter 0]

16 Jun 2023

PONE-D-23-15816Development and validation of a multi-dimensional diagnosis-based comorbidity index that improves prediction of death in men with prostate cancer: Nationwide, population-based register study.PLOS ONE

Dear Dr. Westerberg,

Thank you for submitting your manuscript to PLOS ONE. After careful consideration, we feel that it has merit but does not fully meet PLOS ONE’s publication criteria as it currently stands. Therefore, we invite you to submit a revised version of the manuscript that addresses the points raised during the review process.

We look forward to receiving your revised manuscript.

Kind regards,

Raymond Nienchen Kuo, Ph.D

Academic Editor

PLOS ONE

Journal Requirements:

4. Please note that funding information should not appear in any section or other areas of your manuscript. We will only publish funding information present in the Funding Statement section of the online submission form. Please remove any funding-related text from the manuscript.

  "Funding was received from Swedish Cancer Society (grant number 2022-2051) and Region Uppsala. The funders were not involved in the planning, execution, or completion of the study."

Additional Editor Comments:

The authors employed a nationally-representative prostate cancer database (Prostate Cancer Data Base Sweden version 5 (PCBaSe 5)) to verify the performance of a multidimensional diagnosis-based comorbidity index (i.e., MDCI) on predicting the risk of death. While the findings suggest that MDCI outperforms Charlson Comorbidity Index (CCI) in terms of C-statistics, the authors must address several crucial concerns before this manuscript can be considered suitable for publication.

Reviewers' comments:

Reviewer's Responses to Questions

**Comments to the Author**

1. Is the manuscript technically sound, and do the data support the conclusions?

Reviewer #1: Yes

Reviewer #2: Partly

2. Has the statistical analysis been performed appropriately and rigorously? 

Reviewer #1: Yes

Reviewer #2: Yes

3. Have the authors made all data underlying the findings in their manuscript fully available?

Reviewer #1: No

Reviewer #2: Yes

4. Is the manuscript presented in an intelligible fashion and written in standard English?

Reviewer #1: Yes

Reviewer #2: Yes

5. Review Comments to the Author

Reviewer #1: This is a nicely written article. The authors examined over 280,000 men who did not have prostate cancer to come up with a comorbidity index that is enhanced compared to the Charlson. They use this index (or set of indices) to predict one year, five year, and 10 year all-cause mortality. It was then validated in a large cohort of men with prostate cancer.

Study has multiple strengths including a very large data set spanning many years and robust outcome ascertainment. Mortality is looked at over multiple periods of time. The models end up being impressively better in performance than the Charlson.

Major issues:

1. The reference is the Charlson index. While widely used, this is not the best reference and there are multiple enhancements to the Charlson comorbidity index that have shown superiority, including Elixhauser and the Quan modifications. Why were these not added?

2. The authors compare their improved index to the Charlson but it would be valuable to look at multiple components of their enhanced index or disentangle them. Namely, the value of additional diagnoses alone, the value of a longer look back, and the value of adding occurrence, recency, and duration of related hospitalizations. This is relatively easy to do and would allow us to incrementally understand from a health services research perspective what dimension is actually adding value. It's probably a little bit of each of them but this would be important for researchers because different healthcare databases would have access to some but not all of this information.

3. The authors are relatively silent on the fact that other investigators have attempted to examine the value of adding in measures of hospital admission or severity and the results have not been very good. For example, see the recent paper in JAGS (PMID: 36495264).

4. The drug comorbidity index did surprisingly well, whereas multiple other authors have demonstrated actually poor performance of a drug based multimorbidity measure (for example, Soumerai). The authors do not acknowledge these discrepancies nor provide any explanation.

5. Can the authors provide information on accuracy and comprehensiveness of the databases in question. I understand capture of death as well as comorbidities is excellent but this needs to be presented in brief and referenced.

6. It's unclear to me why duration was categorized as total number of days in hospital exceeding 7 or 14 days (line 131). This is quite unusual analytically and needs justification.

Minor points

1. It is unclear if only inpatient ICD 10 codes are used or outpatient. There seems to be inconsistency in the wording that would suggest one thing or another (for example lines 87 to 90). Kindly clarify this.

2. In the abstract, again please clarify if only inpatient versus outpatient codes were used.

3. Line 51 the word old is used and this is inappropriate and potentially ageist. Simply describe the group you are referring to (e.g. age 65 and older) or use adjectival forms such as older.

4. The paper would benefit from a careful read of spelling and grammar issues, as there are a number of small errors. However, overall readability is still high.

5. Figure one is beautifully done but unfortunately it's not very useful for the average reader or the average researcher. I recommend moving this to a supplement. Instead I would replace figure one with a more granular and focused example of information. Pick a common condition, for example heart failure or COPD, and provide a more illustrative example of the two character, 3 character, and five character length specifics and contributions with more text describing the diagnoses. Appendix 2 part 3 was only slightly helpful and more granular, but more details with one or two specific disease examples would be helpful.

6. Are drug prescriptions available on all adults and how far back does this go?

7. Other important limitations include the fact that other comorbidity measures were not included, the study is restricted to a Swedish population, and it may be difficult to reproduce in other jurisdictions because of some of the unique data that are available in this data set.

8. Looking at figure 2, calibration is suboptimal when one gets to predicted survival probabilities of 90% or less. This is not really commented on adequately in the manuscript.

9. I would also recommend the author spend more time discussing Figure 4, in particular the age based analysis. I found that this is a strength but under emphasized.

10. In terms of the checklist, I found section 15 needed expansion in the paper.

Reviewer #2: Comments to the Author:

The authors present original research that is well written. The issue of risk-stratification in prostate cancer is highly relevant both from a clinical perspective and from a database research perspective, as prostate cancer is often not the cause of death for men carrying this diagnosis. The current manuscript is one of several studies that have attempted to use an unbiased approach (in this case with diagnostic codes) to identify predictors of non-cancer mortality (rather than pre-specifying comorbid medical conditions thought to portend high risks of patients competing morality). What this study adds is a more refined approach by using frequency, recency, and duration to estimate the burden of each predictive diagnostic code. However, there are several methodological decisions including primary model choice, cohort selection, event/endpoint, analytical method, and data visualization that limit the utility and interpretability of their findings. For these reasons I recommend resubmission with major revisions.

Major Points:

1. Primary model choice: the authors chose to report primarily the results of a model with predictors from up to 10 years of data prior to the index date. However, the median pre-index date follow-up length was only 8 years (and many claims-based datasets will have similar limitations in longitudinal data). For these reasons, most claims-based oncology comorbidity models have used 1-year pre-diagnosis data. Additionally, the 1-year model outperformed the other models in discrimination in the validation set. The authors should consider reporting the 1-year model as their primary model both from a predictive performance perspective and for improved adoption to other databases.

2. Cohort selection: there are several issues with the choice of cohorts for training and validation:

a. First, the authors used non-cancer patients exclusively for their model training. This presumes that there are no fundamental differences between these two patient populations. However, in order to be diagnosed with prostate cancer, a patient typically must first undergo screening and then biopsy. Physicians often do not screen men with higher burdens of medical comorbidity (who are less likely to benefit from prostate cancer screening) and men with less healthcare-seeking behavior may have less contact with preventative medicine services such as cancer screenings. Thus the applicability of this predictive model may not be ideal for designing a predictive model for competing morality in prostate cancer patients.

b. The choice of a test dataset based on year of index date is less ideal than using randomization to assign patients to training/test datasets as trends in treatment and screening may have a time-dependent effect.

c. In the context of the above, the authors may still consider using a non-cancer validation cohort to estimate the risk of misclassification of cause-of-death.

3. Study endpoint: the authors chose all-cause mortality as the endpoint for their study. Presumably, they did this because of the inclusion of comparator men (not diagnosed with prostate cancer) who are not at risk of prostate cancer mortality. However, conceptually the interest of a comorbidity index is to assess the likelihood of a given patient of dying from something other than their cancer diagnosis (i.e. their competing morality risk). In using all-cause mortality, the authors lose the ability to stratify between cancer and non-cancer mortality.

4. Analytical method: the choice of survival analysis with Cox regression is suboptimal for the use of a predictive model of comorbidity. This uses a single endpoint (namely death), while fundamentally a comorbidity index in patients newly diagnosed with cancer should describe non-cancer death exclusively (separately from cancer-related death). There are other techniques such as Fine-Gray regression that would allow the authors to correct for this. Given that the comparator men are presumably not at risk of dying from prostate cancer, implementation of a competing risks model should be done in the context of restricting the training and test datasets to men diagnosed with prostate cancer (and omitting entirely the men not diagnosed with cancer as discussed above). Alternatively, if the incidence of prostate cancer specific death is low, the authors could perform survival analysis specifically for non-cancer death.

5. Data visualization: the authors present model calibration curves grouping patients by risk decile and plotting the predicted risk against the 1:1 line. Given the overall favorable prognosis of their patients, this result in a clustering in the upper-right quadrant of the plot, and the authors thus abridge the x-axis. However, what is most useful in a comorbidity index for prostate cancer is to identify the patients most likely to die from their competing comorbid conditions (and thus who may not receive treatment, or possibly undergo a less intensive treatment). For these reasons, the authors should consider grouping patients by risk intervals to show how accurately the model performs at identifying high-risk patients. Indeed, currently the plots suggests excellent performance in the range of low-risk patients but inconsistent performance in the higher-risk range.

Minor Points

1. The authors describe in their methods calculating the Charlson Comorbidity Index (CCI) and then adapting this to their cohort by removing prostate cancer specific diagnostic codes. There is already a standardized and validated CCI for claims-based oncology research (Klabunde et al, PMID: 17531502) including prostate cancer patients that the authors should use for this calculation.

2. The language of “follow-up” as used by the authors references time enrolled in the database prior to the index date, while classically “follow up” in longitudinal cancer survival analyses is with regards to time following the index date. The authors should consider changing this language for clarity.

3. The figure legends appear intercalated in the text body.

4. The authors discuss at length the MDCI’s added value compared to the CCI and the DCI, but do not discuss the added gain of MDCI compared to existing, more simplistic claims-based models (e.g. PMID: 30830794) which require fewer steps in data processing.

5. In the supplements, the authors present the results of their elastic net regression predictors. For user interpretation, it may be reasonable to do the following:

a. Label the diagnostic codes

b. Order them by decreasing order absolute magnitude so that the most impactful predictors are listed first

c. Provide a measure of robustness of the estimate. While elastic net is not amenable to p-values or confidence intervals, it is possible for instance boostrap the regression and show what percentage of the time the same predictors would be selected.

6. PLOS authors have the option to publish the peer review history of their article (what does this mean?). If published, this will include your full peer review and any attached files.

Reviewer #1: No

Reviewer #2: No

---

## [Author Response · Author response to Decision Letter 0]

2 Nov 2023

Journal Requirements

Answer: the manuscript has been updated to meet style requirements.

Answer: The need for consent was waived by The Swedish Ethical Review Authority that approved the project. In NPCR, an opt-out version of consent is applied, i.e. the men are informed in writing and verbally by health care staff, and in addition there is written information displayed on posters in waiting rooms and on the register’s website www.npcr.se. Men who decline to participate are not included in NPCR. 

Subsequentially, a study file was created by linkages of data from NPCR with data in registers held at Statistics Sweden, and the National Board of Health and Welfare by of the Swedish person identity number that was subsequently was replaced by a code. This means that the data set is pseudonymized and due to the large number of variables in the study file it is still considered not anonymized. Therefor the following restrictions apply: we are not allowed to share data on individuals with other researchers, nor or we allowed to upload such data on an open server. However, we can provide access to a study file on a remote server on demand. On the Research platform, data can be uploaded and then accessed by external researchers. However, no individual data are allowed to leave the platform but aggregated data in the form of figures and tables can be exported. This research project has been approved by the Research Ethics Authority (2020-03437) with Professor Pär Stattin as contact person. 

This has been clarified in the Methods section of the manuscript with the following statement (new text bold and underlined):

“The Swedish Ethical Review Authority approved of the study [220-03437 and the need for consent was waiwed. Data was pseudonymized by the National Board of Health and Welfare prior to being delivered to the researchers.” 

Answer: The code will be made available as per your guidelines upon publication. The Data Availability Statement will be updated with a link (DOI) to the publicly availably code hosted at www.zenodo.org under open access license (Creative Commons Attribution 4.0 International [CC BY 4.0]).

4. Please note that funding information should not appear in any section or other areas of your manuscript. We will only publish funding information present in the Funding Statement section of the online submission form. Please remove any funding-related text from the manuscript.

 "Funding was received from Swedish Cancer Society (grant number 2022-2051) and Region Uppsala. The funders were not involved in the planning, execution, or completion of the study."

Answer: Funding has been removed from the manuscript.

Answer: We have revised the cover letter accordingly and made additional clarifications (also see answer to point 2 above). The data availability statement has been updated and is formulated the same as the Data Availability Statements of two recent publications in PLOS ONE based on PCBaSe data authored by some of the coauthors of this manuscript. We additionally provided an additional, non-author point of contact in bold. It now reads: 

“Data used in the present study was extracted from the Prostate Cancer Database Sweden (PCBaSe), which is based on the National Prostate Cancer Register (NPCR) of Sweden and linkage to several national health-data registers. The data cannot be shared publicly because the individual-level data contain potentially identifying and sensitive patient information and cannot be published due to legislation and ethical approval (https://etikprovningsmyndigheten.se). Use of the data from national health-data registers is further restricted by the Swedish Board of Health and Welfare (https://www.socialstyrelsen.se/en/) and Statistics Sweden (https://www.scb.se/en/) which are Government Agencies providing access to the linked healthcare registers. The data will be shared on reasonable request in an application made to any of the steering groups of NPCR and PCBaSe (contact npcr@npcr.se). For detailed information, please see www.npcr.se/in-english, where registration forms, manuals, and annual reports from NPCR are available alongside a full list of publications from PCBaSe. The code used for the analyses can be accessed via the following link (DOI will be inserted).”

Answer: The manuscript has been updated accordingly.

Additional Editor Comments:

The authors employed a nationally-representative prostate cancer database (Prostate Cancer Data Base Sweden version 5 (PCBaSe 5)) to verify the performance of a multidimensional diagnosis-based comorbidity index (i.e., MDCI) on predicting the risk of death. While the findings suggest that MDCI outperforms Charlson Comorbidity Index (CCI) in terms of C-statistics, the authors must address several crucial concerns before this manuscript can be considered suitable for publication.

Reviewers' comments - Reviewer #1 

Major issues:

1. The reference is the Charlson index. While widely used, this is not the best reference and there are multiple enhancements to the Charlson comorbidity index that have shown superiority, including Elixhauser and the Quan modifications. Why were these not added?

Answer: We acknowledge the reviewer’s comment and agree that this issue needs further clarification in the manuscript. The definition of the Charlson comorbidity index used in our analyses were those most recently recommended for use with the Swedish Patient Register. A reference for this (Ludvigsson JF et al. Adaptation of the Charlson Comorbidity Index for Register-Based Research in Sweden. Clinical epidemiology. 2021;13:21-41. Epub 2021/01/21. doi: 10.2147/CLEP.S282475. PubMed PMID: 33469380; PubMed Central PMCID: PMCPMC7812935) is provided in the methods section. 

We fully acknowledge that there are a number of other adaptations of the Charlson index, as well as other approaches to summarize information on comorbidity from diagnosis codes, such as the Elixhauser index. It is outside the scope of this study to make a comparison between available indices. We chose the Charlson index as a comparator because of its widespread use and scientific interest. A PubMed search restricted to the period 2005 to present date supports this selection:

Charlson ("Charlson index" OR "Charlson comorbidity index") N=10,950

Elixhauser ("Elixhauser index" OR " Elixhauser comorbidity index") N=590

The Quan modification of the Charlson Comorbidity Index to administrative data is older and not specifically optimized to Swedish data. We argue that it is best to use the optimal version of the comparator, in order not to overestimate the relative performance of our new comorbidity index. We have, however, also added a statement in the limitations section (new text bold and underlined):

“Comparisons to other comorbidity measures, including other adaptations of the Charlson comorbidity index and the Elixhauser comorbidity index, may yield different results.“

2. The authors compare their improved index to the Charlson but it would be valuable to look at multiple components of their enhanced index or disentangle them. Namely, the value of additional diagnoses alone, the value of a longer look back, and the value of adding occurrence, recency, and duration of related hospitalizations. This is relatively easy to do and would allow us to incrementally understand from a health services research perspective what dimension is actually adding value. It's probably a little bit of each of them but this would be important for researchers because different healthcare databases would have access to some but not all of this information.

Answer: Although analyses of other adaptations of the Charlson index is of interest, this is outside of the scope of the present study. Incorporating such an analysis would make the ms far too long. 

Our results clearly indicate the relevance and promising potential of a multi-dimensional strategy for optimization of comorbidity measures, but further evaluation and development should best be done in a broader and more general population, notably e.g. also including women. This is already a planned research project, using the entire Swedish population since 1998. The different dimensions of comorbidity relative to different cancer diagnoses can then be assessed comprehensively. Although in principle the analyses may seem easy, it would also take a long time to generate these results since there are a number of different factors to investigate, the analyses have to be carefully pre-planned, and the cross-validation procedure is computationally intensive. It is appropriate to undertake this in a larger and more generalizable population. We therefore respectfully would prefer not to pursue this further within this study. 

3. The authors are relatively silent on the fact that other investigators have attempted to examine the value of adding in measures of hospital admission or severity and the results have not been very good. For example, see the recent paper in JAGS (PMID: 36495264).

Answer: The reviewer is thanked for highlighting this. The provided reference was based on Medicare claims linked with Health and Retirement Study data (5012 subjects) and developed 7 markers based on the number of disease-associated outpatient visits, emergency department visits, and hospitalizations of a patient over a defined interval. 

The manuscript has been updated with the following changes in the discussion of the manuscript: 

“Similar attempts have been made. A recent study using Medicare claims linked with Health and Retirement Study data developed 7 dichotomous markers of disease severity, including markers based on the number of disease-associated outpatient visits, emergency department visits, and hospitalizations made by an individual over a defined interval. These markers did not, however, contribute meaningfully to predict outcomes such as decline in activities of daily living decline or mortality.”

4. The drug comorbidity index did surprisingly well, whereas multiple other authors have demonstrated actually poor performance of a drug based multimorbidity measure (for example, Soumerai). The authors do not acknowledge these discrepancies nor provide any explanation.

Answer: We respectfully argue that a comprehensive discussion on different drug indices is not within the scope of this study since the present study is focused on creation and evaluation of a multi-dimensional quantitative approach to extract information of comorbidity from ICD-10 diagnosis codes. The drug comorbidity index only serves as a comparator of performance. It is not our aim to evaluate and compare different prescription-based comorbidity indices. In our original publication of the DCI (Gedeborg R et al. An Aggregated Comorbidity Measure Based on History of Filled Drug Prescriptions: Development and Evaluation in Two Separate Cohorts. Epidemiology 2021;32: 607-15. PMID:33935137) we compare its performance with another recently developed drug index. 

5. Can the authors provide information on accuracy and comprehensiveness of the databases in question. I understand capture of death as well as comorbidities is excellent but this needs to be presented in brief and referenced.

Answer: The manuscript has been updated with a statement about the quality of the key databases. 

The following has been added to the Materials section (new text bold and underlined):

“The capture rate of the National Prostate Cancer Register is above 96% compared to the Swedish Cancer register to which reporting is mandated by law [14]. There are only modest differences in demographics, cancer treatment, comorbidity, and mortality between the men in NPCR and men only registered in the Cancer Register, indicating that information in NPCR can be generalized to all men with prostate cancer in Sweden.”

/…/

“All codes according to the 10th revision of International Statistical Classification of Diseases and Related Health Problems (ICD-10) registered as related to hospitalizations or specialist outpatient visits up to 10 years prior to the index date were extracted from the inpatient and specialist outpatient sub-registers of the National Patient Register. The National Patient Register comprises information on all in-hospital care and out-patient specialist care in Sweden. It has nation-wide coverage of in-patient care since 1987 and specialized outpatient care since 2001. During the study period, diagnoses were recorded according to the Swedish clinical modification of ICD-10 which has few modifications compared to the original ICD-10 version. Validation studies indicate that coding accuracy is diagnosis-specific [15-17]. 

 The Swedish Prescribed Drug Register, used in this study to calculate a drug comorbidity index, contains details of all prescriptions dispensed in Sweden since July 1, 2005. Drugs are identified by a unique identifier for each specific combination of brand name, substance, formulation and package. The register only includes filled prescriptions, and not medicines sold over the counter or medicines administered directly to the patient during in-patient care, out-patient care or primary care. 

Dates of death until 31 December 2020 were extracted from the Cause of Death Register for the follow-up of mortality. The Swedish cause of death register contains information on all deaths of Swedish residents since 1952.“

6. It's unclear to me why duration was categorized as total number of days in hospital exceeding 7 or 14 days (line 131). This is quite unusual analytically and needs justification.

Answer: We acknowledge that this should be commented in the methods section. The intention is to define indicator variables that separates short observational and mainly due to hospitalization due to less severe disease and well-preserved functional status from patients with severe conditions and/or reduced functional status. It should be recognized that duration of hospitalization is also determined by factors such as bed space availability, and health care patterns. We therefore prespecified two arbitrary cut-offs based on clinical experience and reasoning. Notably, this represents a standard stratification of a continuous variable for length of hospital stay into three categories, using two dummy binary variables. We made no attempt to optimize the cutoffs, in line with our response to question 2 above. 

The manuscript has been updated in the Methods section to clarify the rationale behind these variables (new text bold and underlined):

“Duration of hospital admission was categorized by dummy variables indicating if the total number of days in hospital exceeded 7 or 14 days. These cutoffs were prespecified based on clinical reasoning with the intention to separate patients with short observational and mainly diagnostic stays from patients with severe conditions and/or reduced functional status.”

Minor points

1. It is unclear if only inpatient ICD 10 codes are used or outpatient. There seems to be inconsistency in the wording that would suggest one thing or another (for example lines 87 to 90). Kindly clarify this.

Answer: We thank the reviewer for noticing this. Both inpatient and specialist outpatient ICD-10 codes were used. This is stated in the Materials subsection of the Material and Methods section and has now also been clarified in the Abstract and Methods subsection.

2. In the abstract, again please clarify if only inpatient versus outpatient codes were used.

Answer: The abstract has been updated accordingly (see previous comment).

3. Line 51 the word old is used and this is inappropriate and potentially ageist. Simply describe the group you are referring to (e.g. age 65 and older) or use adjectival forms such as older.

Answer: This is fully agreed (especially by the “older” coauthors). The manuscript has been updated accordingly.

4. The paper would benefit from a careful read of spelling and grammar issues, as there are a number of small errors. However, overall readability is still high.

Answer: The reviewer is thanked for noticing this and manuscript has been revised for language. 

5. Figure one is beautifully done but unfortunately it's not very useful for the average reader or the average researcher. I recommend moving this to a supplement. Instead I would replace figure one with a more granular and focused example of information. Pick a common condition, for example heart failure or COPD, and provide a more illustrative example of the two character, 3 character, and five character length specifics and contributions with more text describing the diagnoses. Appendix 2 part 3 was only slightly helpful and more granular, but more details with one or two specific disease examples would be helpful.

Answer: We agree and have improved Fig 1 according to the suggestions, see below. It now illustrates contributions with a specific example of codes under diagnosis I5 (Heart failure and some heart disorders and diseases) and of all diagnoses under I (cardiovascular diseases). 

6. Are drug prescriptions available on all adults and how far back does this go?

Answer: Yes. The Prescribed Drug Register started to record drug prescriptions in July 2005 and data is available from this date and onward. The drug comorbidity index was developed using a 1-year look-back in PMID: 33935137 and this is what was used in the current manuscript. 

7. Other important limitations include the fact that other comorbidity measures were not included, the study is restricted to a Swedish population, and it may be difficult to reproduce in other jurisdictions because of some of the unique data that are available in this data set.

Answer: This is agreed and the Discussion has been updated accordingly. We do, however, find it important not to restrict methodological development because of lower availability of data in other settings. The overall aim should be to optimize epidemiological methods to minimize bias. This includes development of both data quality and analytical methods. 

The manuscript has been updated in the Discussion section to acknowledge these potential limitations (new text bold and underlined):

“There are some limitations to our study. Linkages of rich sources at an individual level can be done in Sweden but not everywhere else so this is a limitation for generalization of this method to other countries and settings. ”

8. Looking at figure 2, calibration is suboptimal when one gets to predicted survival probabilities of 90% or less. This is not really commented on adequately in the manuscript.

Answer: This is agreed and the manuscript has been revised accordingly. The key problem is the substantial uncertainty in the estimates in the lower probability range, due to the limited availability of data in that region. We have tried to provide a more informative analysis of calibration with a revised plot, where we have added CCI for contextualization and also indicate the number of observations in each strata in a table below the figure. This also nicely depicts that the MDCI provides better separation of survival probabilities than the CCI, which likely contributes to better predictive performance.

The manuscript has been updated in the Results section with a new Fig. 2 and expanded text (new text bold and underlined):

“Calibration curves comparing predicted and observed 10-year probabilities of death indicated good calibration of the MDCI, and also illustrates the increased separation provided by the MDCI compared to the CCI (Fig. 2). Calibration was similar for predicted 5-year mortality risk but less optimal for prediction of 1-year mortality, where the MDCI tended to underestimate mortality risk in stata with low mortality and overestimate the risk in strata with higher mortality risk (Fig. S4).”

9. I would also recommend the author spend more time discussing Figure 4, in particular the age based analysis. I found that this is a strength but under emphasized.

Answer: The Discussion section of the manuscript has been updated accordingly (new text bold and underlined):

“Our new multidimensional diagnosis-based comorbidity index (MDCI) based on occurrence, recency, frequency, and duration of hospital admissions for all ICD-10 codes predicted risk of death better than the commonly used Charlson comorbidity index. This was demonstrated consistently across strata of both age and other comorbidity indices. Notably, the MDCI was also able to stratify men with Charlson comorbidity index = 0 into strata with clearly different risk of death. This supports the hypothesis that relevant quantitative information related to disease severity can be extracted from diagnosis codes in longitudinal register data.”

10. In terms of the checklist, I found section 15 needed expansion in the paper.

Answer: The comment refers to the item “Model specification”. The full prediction model does not fit in the main article because there are so many predictors, and is presented in the supplement (all coefficients). We have expanded the text related to “Model specification” to further clarify how the model was specified and how it can be used (new text bold and underlined): 

To address this the registered ICD-10 codes were first processed in a data management step involving code truncation and elongation, pruning of unnecessary codes, and filtering codes present in at least 0.01% of the development cohort (Fig. S1 and S2 Appendix). 

/…/

To compute a patient’s MDCI, one sums the coefficients from this model related to the selected predictors based on the ICD-10 codes observed for each patient during the 10 years preceding the index date.

/…/

Ten-fold cross-validation over a grid of 100 values of the hyperparameter that controls the amount of penalization was used to identify the model with the highest concordance index (c-index).

Reviewer #2: Comments to the Author:

Major Points:

1. Primary model choice: the authors chose to report primarily the results of a model with predictors from up to 10 years of data prior to the index date. However, the median pre-index date follow-up length was only 8 years (and many claims-based datasets will have similar limitations in longitudinal data). For these reasons, most claims-based oncology comorbidity models have used 1-year pre-diagnosis data. Additionally, the 1-year model outperformed the other models in discrimination in the validation set. The authors should consider reporting the 1-year model as their primary model both from a predictive performance perspective and for improved adoption to other databases.

Answer: The comment is relevant and it is agreed that further clarification is needed. 

ICD-10 codes to be used for risk prediction were available for all subjects during the 10 years prior to each subject’s index date (i.e. pre-index date data). “Follow-up” refers to the period after the index date until last date of follow-up (2021-12-31) or date of death. The median length of follow-up after the index date was 8 years. We have now tried to clarify this distinction throughout the manuscript. 

Regarding model performance: The 1-year model did not outperform the other models in the validation datasets. In fact, performance improved with increasing length of the follow-up period, both in terms of discrimination and calibration. In terms of performance, all three models were evaluated after 1, 5 and 10 years of follow-up for mortality since the c-index is a time-dependent measure. Within each combination of dataset and length of follow-up for mortality used for computing the c-index, the model developed using a 10-year follow-up period performed best. We believe that the labeling in Table 2 may have caused this confusion and have clarified this in the revised manuscript both in Table 2 and throughout the manuscript. For these reasons, we choose to report the results of the 10-year model as the primary model. We now also stressed this observation in the summary of main results in the Discussion section (new text bold and underlined):

“The prediction model developed using 10 years of follow-up performed better than models based on shorter follow-up.”

2. Cohort selection: there are several issues with the choice of cohorts for training and validation:

a. First, the authors used non-cancer patients exclusively for their model training. This presumes that there are no fundamental differences between these two patient populations. However, in order to be diagnosed with prostate cancer, a patient typically must first undergo screening and then biopsy. Physicians often do not screen men with higher burdens of medical comorbidity (who are less likely to benefit from prostate cancer screening) and men with less healthcare-seeking behavior may have less contact with preventative medicine services such as cancer screenings. Thus the applicability of this predictive model may not be ideal for designing a predictive model for competing morality in prostate cancer patients.

Answer: We agree that men diagnosed with prostate cancer may constitute a selected subgroup of all men, also in terms of comorbidity. However, we do not agree that prostate cancer patients should be included in model development. The purpose of a baseline comorbidity measure is to indicate the mortality risk from comorbidity, had the individual not had the primary condition (here prostate cancer). Baseline mortality risk from the prostate cancer itself require specific and detailed information on the prostate cancer at diagnosis.

If prostate cancer patients had been included it is likely that predictive ability would increase, but generalizability would decrease. From this perspective our results are likely conservative, which is appropriate. Notably, if the purpose of a comorbidity measure is to only adjust for comorbidity in an analysis, it is likely most effective to derive the comorbidity measure internally in the study population, but this would require a very extensive dataset. 

We do, however, fully agree that it is most important to provide validation in a prostate cancer population, and this is the reason for having two separate validation cohorts, one with and one without prostate cancer. 

The Discussion (limitations) section of the manuscript has been updated to reflect the selection of the development dataset (new text bold and underlined):

“The prediction models were developed in men selected as comparison men to men with prostate cancer, meaning that e.g. extrapolation of model parameters to women or specific other patient groups should be cautious.”

b. The choice of a test dataset based on year of index date is less ideal than using randomization to assign patients to training/test datasets as trends in treatment and screening may have a time-dependent effect.

Answer: We respectfully disagree regarding this issue. Randomization would yield very similar development (training) and validation (test) cohorts (men without prostate cancer) and any differences between these two cohorts in terms of discrimination (e.g. c-index) would be mainly due to chance alone (as long as we are not overfitting, and the c-index varied very little in the cross-validation across subsets of the development data and in the validation data which suggests that there was no apparent overfitting). A test dataset derived using a random process would essentially be a mirror image of the development dataset.

Validation in a test set derived using random splitting of the study data does therefore not reduce the risk for overly optimistic estimation of predictive performance to be expected in other data sources. 

Using a temporal validation, as in our study, comes closer to an external validation, since we hope that the model will be relevant also for future patients (Ramspek CL, Jager KJ, Dekker FW, Zoccali C, van Diepen M. External validation of prognostic models: what, why, how, when and where? Clinical Kidney Journal 2020;14: 49-58).

We fully agree that time trends in treatment and screening may have an impact on model performance and the temporal validation is therefore a conservative evaluation of model performance, and provides a more balanced evaluation for assessment of potential generalizability. 

We have added a clarifying statement in the Methods section (new text bold and underlined):

”Temporal rather than random splitting of the data is expected to reduce the risk for overly optimistic estimates of model performance.”

c. In the context of the above, the authors may still consider using a non-cancer validation cohort to estimate the risk of misclassification of cause-of-death.

Answer: We agree that the validation cohort without prostate cancer is relevant.

3. Study endpoint: the authors chose all-cause mortality as the endpoint for their study. Presumably, they did this because of the inclusion of comparator men (not diagnosed with prostate cancer) who are not at risk of prostate cancer mortality. However, conceptually the interest of a comorbidity index is to assess the likelihood of a given patient of dying from something other than their cancer diagnosis (i.e. their competing morality risk). In using all-cause mortality, the authors lose the ability to stratify between cancer and non-cancer mortality.

Answer: The concern regarding cause-specific death is relevant. Notably, comparator men are also risk of cancer death. They can have any other cancer at the index date but not prostate cancer at index date and they can be diagnosed with prostate cancer after the index date. 

To evaluate how comorbidity impacts mortality, it is for several reasons appropriate to first evaluate impact on all-cause mortality. We agree with the reviewer that cause-specific mortality may be of particular relevance in a specific patient population. But then all-cause mortality must be handled as a competing risk and misclassification of cause of death must be considered. Misclassification of prostate cancer death has been shown to be an important problem:

1. Orrason AW, Styrke J, Garmo H, Stattin P. Evidence of cancer progression as the cause of death in men with prostate cancer in Sweden. BJU Int 2023;131: 486-93. PMID:36088648.

2. Innos K, Paapsi K, Alas I, Baum P, Kivi M, Kovtun M, Okas R, Pokker H, Rajevskaja O, Rautio A, Saretok M, Valk E, Žarkovski M, Denissov G, Lang K. Evidence of overestimating prostate cancer mortality in Estonia: a population-based study. Scand J Urol 2022;56: 359-64. PMID:36073064.

3. Löffeler S, Halland A, Weedon-Fekjær H, Nikitenko A, Ellingsen CL, Haug ES. High Norwegian prostate cancer mortality: evidence of over-reporting. Scand J Urol 2018;52: 122-8. PMID:29325479.

4. Turner EL, Metcalfe C, Donovan JL, Noble S, Sterne JA, Lane JA, E IW, Hill EM, Down L, Ben-Shlomo Y, Oliver SE, Evans S, Brindle P, Williams NJ, Hughes LJ, Davies CF, Ng SY, Neal DE, Hamdy FC, Albertsen P, Reid CM, Oxley J, McFarlane J, Robinson MC, Adolfsson J, Zietman A, Baum M, Koupparis A, Martin RM. Contemporary accuracy of death certificates for coding prostate cancer as a cause of death: Is reliance on death certification good enough? A comparison with blinded review by an independent cause of death evaluation committee. Br J Cancer 2016;115: 90-4. PMID:27253172.

5. Fall K, Strömberg F, Rosell J, Andrèn O, Varenhorst E. Reliability of death certificates in prostate cancer patients. Scand J Urol Nephrol 2008;42: 352-7. PMID:18609293.

6. Hoffman RM, Stone SN, Hunt WC, Key CR, Gilliland FD. Effects of misattribution in assigning cause of death on prostate cancer mortality rates. Ann Epidemiol 2003;13: 450-4. PMID:12875804.

Evaluation of cause-specific mortality is consequently relevant but complex, and it does not fit within the scope of the present study. Please see response to Reviewer #1 question 2 regarding planned continuation of the research program. 

4. Analytical method: the choice of survival analysis with Cox regression is suboptimal for the use of a predictive model of comorbidity. This uses a single endpoint (namely death), while fundamentally a comorbidity index in patients newly diagnosed with cancer should describe non-cancer death exclusively (separately from cancer-related death). There are other techniques such as Fine-Gray regression that would allow the authors to correct for this. Given that the comparator men are presumably not at risk of dying from prostate cancer, implementation of a competing risks model should be done in the context of restricting the training and test datasets to men diagnosed with prostate cancer (and omitting entirely the men not diagnosed with cancer as discussed above). Alternatively, if the incidence of prostate cancer specific death is low, the authors could perform survival analysis specifically for non-cancer death.

Answer: We appreciate the comments regarding alternative study designs and analytical approaches. This stimulates the scientific discussion for the planning of future studies. Regarding prostate-cancer specific mortality, please see the response to the previous question. A mature analysis of prostate-cancer specific mortality in a general prostate cancer population requires a follow-up of 30-40 years. It may be more clinically relevant and feasible to focus on a subpopulation, such as men with castration-resistant prostate cancer. We have performed such an evaluation of our drug comorbidity index (please see Fallara G, Gedeborg R, Bill-Axelson A, Garmo H, Stattin P. A drug comorbidity index to predict mortality in men with castration resistant prostate cancer. PLoS One 2021;16: e0255239. PMID:34320037). The complexity of such analyses does not fit within the scope of our present aim but requires a separate study. Our present study provides proof-of-concept for such an undertaking. 

5. Data visualization: the authors present model calibration curves grouping patients by risk decile and plotting the predicted risk against the 1:1 line. Given the overall favorable prognosis of their patients, this result in a clustering in the upper-right quadrant of the plot, and the authors thus abridge the x-axis. However, what is most useful in a comorbidity index for prostate cancer is to identify the patients most likely to die from their competing comorbid conditions (and thus who may not receive treatment, or possibly undergo a less intensive treatment). For these reasons, the authors should consider grouping patients by risk intervals to show how accurately the model performs at identifying high-risk patients. Indeed, currently the plots suggests excellent performance in the range of low-risk patients but inconsistent performance in the higher-risk range.

Answer: The comment is relevant and the calibration plots have been revised to improve clarity. Please see the response to Reviewer #1 question 8 which is also related to this concern. 

Minor Points

1. The authors describe in their methods calculating the Charlson Comorbidity Index (CCI) and then adapting this to their cohort by removing prostate cancer specific diagnostic codes. There is already a standardized and validated CCI for claims-based oncology research (Klabunde et al, PMID: 17531502) including prostate cancer patients that the authors should use for this calculation.

Answer: It is agreed that the CCI should be calculated using an optimized algorithm, so that the comparison is as favorable as possible for the CCI. The exclusion of ICD-10 codes for prostate cancer was only done when CCI was calculated in the validation cohort with prostate cancer cases. 

• The study by Klabunde et al is focused on cancer populations and is therefore only relevant for the calculation of CCI in our validation cohort with prostate cancer patients. 

• The key new feature of CCI as calculated by Klabunde et al is the use of both Medicare inpatient and physician claims. This is in line with our use of both inhospital and specialist outpatient data. 

• Klabunde et al used the comorbidity categories as originally described by Charlson et al. This requires adaptation to ICD-9-CM codes in the Klabunde et al study, and in our case to ICD-10-SE (Swedish clinical modification of ICD-10). We have used the mapping recently suggested as optimal for ICD-10 codes in the Swedish National Patient Register (Ludvigsson et al. Adaptation of the Charlson Comorbidity Index for Register-Based Research in Sweden. Clinical epidemiology 2021;13: 21-41. PMID:33469380). This should provide optimal mapping for our data.

• Klabunde et al excluded the diagnostic codes corresponding to solid tumors and leukemia/lymphoma. We do not agree to this approach since patients with prostate cancer can have other malignancies in their history, and they are likely relevant for the burden of comorbidity at baseline. We excluded only codes related to prostate cancer. 

The approach used in our study to calculate CCI in the validation cohort with prostate cancer is consequently to the extent possible entirely in line with the approach used in Klabunde et al. We have therefore added this reference to the methods section (new text bold and underlined):

“When CCI was calculated in the validation cohort with prostate cancer cases we excluded ICD-10 codes for prostate cancer (C61), and metastases (C77-80) if they were registered in conjunction with C61. This is in line with previous adaptations of the CCI to cancer populations [Ref Klabunde et al].”

2. The language of “follow-up” as used by the authors references time enrolled in the database prior to the index date, while classically “follow up” in longitudinal cancer survival analyses is with regards to time following the index date. The authors should consider changing this language for clarity.

Answer: This is fully agreed and the manuscript has been revised to consistently clarify this. 

3. The figure legends appear intercalated in the text body.

Answer: We have revised the manuscript according to the journal guidelines, and in particular noted that “Each figure caption should appear directly after the paragraph in which they are first cited.” 

4. The authors discuss at length the MDCI’s added value compared to the CCI and the DCI, but do not discuss the added gain of MDCI compared to existing, more simplistic claims-based models (e.g. PMID: 30830794) which require fewer steps in data processing.

Answer: Please see our response to Reviewer #1 question 1 and 2. The focus of our study is on the added value of deriving quantitative information from diagnosis codes in longitudinal data before the index date. The study by Riviere et al is highly relevant, uses similar analytical strategies as in our study, but includes also data on demographics, procedures, laboratory data, and cancer severity staging. While we fully agree that these addition types of data should be included in baseline risk prediction, our study aim was focused on optimizing information from ICD-10 diagnosis codes. We therefore did not compare with predictions including other types of data. 

5. In the supplements, the authors present the results of their elastic net regression predictors. For user interpretation, it may be reasonable to do the following:

a. Label the diagnostic codes

b. Order them by decreasing order absolute magnitude so that the most impactful predictors are listed first

c. Provide a measure of robustness of the estimate. While elastic net is not amenable to p-values or confidence intervals, it is possible for instance boostrap the regression and show what percentage of the time the same predictors would be selected.

Answer: We appreciate the thoughtful suggestion. 

a. We have labeled the diagnostic codes (based on the first two code-characters)

b. We ordered the coefficients as suggested

c. We fully agree that it is of interest to evaluate the relative contribution of individual predictors, or rather which quantitative aspects of health-care utilization that are most predictive. This is an extensive expansion of the study objective and is therefore not possible to report within the current study. It is being pursued in a separate study program, which will utilize the entire Swedish population since 1998, and therefore allow in-depth analyses of which aspects of the patient history and which variables are most predictive, in the general population as well as in various cancer populations. We have therefore added a limitation in the Discussion to acknowledge this issue (new text bold and underlined): 

“Future research should assess the discrimination and calibration of the MDCI in the general population, including both men and women. In such a general population sample, it is of interest to try to identify what specific features of this new index that drives the improved predictive ability, so that the principle also can be adapted to other settings with less comprehensive data sources.”

---

## [Decision Letter · Decision Letter 1]

14 Nov 2023

PONE-D-23-15816R1Development and validation of a multi-dimensional diagnosis-based comorbidity index that improves prediction of death in men with prostate cancer: Nationwide, population-based register study.PLOS ONE

Dear Dr. Westerberg,

Thank you for submitting your manuscript to PLOS ONE. After careful consideration, we feel that it has merit but does not fully meet PLOS ONE’s publication criteria as it currently stands. Therefore, we invite you to submit a revised version of the manuscript that addresses the points raised during the review process.

The manuscript has indeed undergone significant enhancements during the course of the review. Nonetheless, the second reviewer has identified a few minor issues that remain to be addressed. These issues, while minor, are crucial to the manuscript's completeness and therefore, warrant your attention before we can endorse the manuscript for publication.

We look forward to receiving your revised manuscript.

Kind regards,

Raymond Nienchen Kuo, Ph.D

Academic Editor

PLOS ONE

Journal Requirements:

Reviewers' comments:

Reviewer's Responses to Questions

**Comments to the Author**

1. If the authors have adequately addressed your comments raised in a previous round of review and you feel that this manuscript is now acceptable for publication, you may indicate that here to bypass the “Comments to the Author” section, enter your conflict of interest statement in the “Confidential to Editor” section, and submit your "Accept" recommendation.

Reviewer #1: (No Response)

Reviewer #2: All comments have been addressed

2. Is the manuscript technically sound, and do the data support the conclusions?

Reviewer #1: Yes

Reviewer #2: Yes

3. Has the statistical analysis been performed appropriately and rigorously? 

Reviewer #1: Yes

Reviewer #2: Yes

4. Have the authors made all data underlying the findings in their manuscript fully available?

Reviewer #1: Yes

Reviewer #2: No

5. Is the manuscript presented in an intelligible fashion and written in standard English?

Reviewer #1: Yes

Reviewer #2: Yes

6. Review Comments to the Author

Reviewer #1: The authors have done a very nice job responding to almost all of my comments. I appreciate the additional details and explanations. The revised manuscript and supplemental materials are substantially improved and nearly ready for publication. In my view only 4 things remain to be done.

Most importantly, I respectfully disagree with the authors' response to my suggestion to disaggregate their index's performance to aid other scientists in understanding whether the improved predictive ability is due to a longer lookback, addition of medications, or addition of comorbidities. This is actually one of the key insights of this paper, even recognizing the limitations and challenges the authors provide. It is not difficult to do an illustrative analysis rather than a comprehensive analysis. For example, for the outcome of 10-year overall survival, show the performance of the CCI, then extend lookback, then add additional conditions, then add medications. Report the C-statistic for each step. No need for external validation.

Second, while I like revised Figure 1, it would be helpful to have English labels/diagnoses for each level of characters, e.g. I5, I51, I500, to allow readers to better understand the level of specificity being illustrated.

Third, with at least 16 supplemental materials, I find this incredibly comprehensive but excessive. Some things can be combined or removed. For example, table S6 and S7 can be combined. Tables S1 and S4 could be removed in my view or made available upon writing to the authors.

Finally, figure S4 would benefit from a blow-out of the predicted survival in the 0.8-1.0 or 0.9-1.0 range, where 'most of the action is'. This could either be done using a non-linear graph or a graph within graph, at least for the 3 development cohort figures (otherwise we have 9 graphs to revise).

Reviewer #2: The authors have made improvements to the figures/tables to improve the interpretability of their work. They provide reasonable explanation for the scope and aim of their work in response to some methodological critiques, especially in light of their ongoing follow-up studies within this domain.

7. PLOS authors have the option to publish the peer review history of their article (what does this mean?). If published, this will include your full peer review and any attached files.

Reviewer #1: No

Reviewer #2: No

---

## [Author Response · Author response to Decision Letter 1]

4 Dec 2023

See attached file "Answers to reviewers" for figures and tables and correct formatting of text.

Reviewer #1: The authors have done a very nice job responding to almost all of my comments. I appreciate the additional details and explanations. The revised manuscript and supplemental materials are substantially improved and nearly ready for publication. In my view only 4 things remain to be done.

We thank the Reviewer for the constructive comments that we have attempted to fully address below. 

Most importantly, I respectfully disagree with the authors' response to my suggestion to disaggregate their index's performance to aid other scientists in understanding whether the improved predictive ability is due to a longer lookback, addition of medications, or addition of comorbidities. This is actually one of the key insights of this paper, even recognizing the limitations and challenges the authors provide. It is not difficult to do an illustrative analysis rather than a comprehensive analysis. For example, for the outcome of 10-year overall survival, show the performance of the CCI, then extend lookback, then add additional conditions, then add medications. Report the C-statistic for each step. No need for external validation.

Answer: We thank the reviewer for the suggestions and have made complementary analyses based on the development dataset in line with the suggestions. 

We first assessed the impact of lookback on the CCI and/or in combination with the DCI (drug comorbidity index) (see table below). We found that a longer lookback increases the C-index, in particular when CCI is used alone, and that the C-index increases when CCI is combined with DCI, in which case lookback has some, but less, impact. 

Lookback for CCI (years) Variables C-index 95% CI

1 CCI 0.599 (0.597 - 0.601)

1 CCI and DCI 0.715 (0.713 - 0.717)

5 CCI 0.654 (0.652 - 0.655)

5 CCI and DCI 0.721 (0.719 - 0.723)

10 CCI 0.668 (0.666 - 0.670)

10 CCI and DCI 0.724 (0.722 - 0.725)

Note that the DCI has a fixed lookback of 1 year.

We also similarly assessed the impact of lookback for the MDCI (see table below). The findings were comparable to the above although the C-indices were slightly larger overall. In particular, MDCI using a lookback of 10 years performed better than CCI and DCI combined, while MDCI and DCI combined performed the best. 

Lookback for MDCI (years) Variables C-index 95% CI

1 MDCI 0.637 (0.635 - 0.639)

1 MDCI and DCI 0.722 (0.720 - 0.724)

5 MDCI 0.716 (0.714 - 0.718)

5 MDCI and DCI 0.743 (0.742 - 0.745)

10 MDCI 0.734 (0.732 - 0.735)

10 MDCI and DCI 0.751 (0.749 - 0.752)

Note that the DCI has a fixed lookback of 1 year.

The above findings have been summarized in a new Supplementary Table 3, and we have made the following comment in the methods section: 

“In an exploratory analysis using the development dataset we also evaluated the impact of length of lookback period for both the Charlson comorbidity index and the MDCI, alone or in combination with the drug comorbidity index.”

We have made the following comments in the results section: 

“In complementary analyses, decreasing the length of the lookback period to 5 years or 1 year for the Charlson comorbidity index or for the MDCI clearly decreased the predictive ability, e.g. using a lookback of one year the C-index for the Charlson comorbidity index and MDCI were 0.599 (95% CI: 0.597 - 0.601) and 0.637 (0.635 - 0.639), respectively (Supplementary Table S3). When the comorbidity indices were combined with the drug comorbidity index the overall predictive ability increased and the impact of lookback was less pronounced, although a lookback of 10 years produced the largest C-indices both for Charlson comorbidity index and the MDCI, 0.724 (95% CI: 0.722 - 0.725) and 0.751 (95% CI: 0.749 - 0.752), respectively (Supplementary Table S3).”

We have made the following changes in the discussion (new text in bold): 

“Our results suggest that the MDCI and DCI reflect different aspects of a patient’s medical history and therefore should be used in combination, preferably with a lookback of 5-10 years for the MDCI.”

and

“While the study population was large, the results may be influenced by the restriction to men and the distribution of comorbidities in this particular population. The study population was therefore not well suited to elucidate the relative contribution of predictors based on occurrence, frequency, duration, and recency to the overall performance of the MDCI. The main purpose of this study was to provide proof of principle for a multidimensional diagnosis-based comorbidity index.” 

Second, while I like revised Figure 1, it would be helpful to have English labels/diagnoses for each level of characters, e.g. I5, I51, I500, to allow readers to better understand the level of specificity being illustrated.

Answer: We thank the reviewer for the suggestion. We have improved the figure based on the suggestion and updated the figure text (see below). 

Updated figure text (new text in bold): 

“Figure 1. Example of predictors selected in the prediction model. ICD-10 codes for cardiovascular diseases are shown in the figure, specifically focusing on the ICD-10 code category I5 (heart failure and some heart disorders and diseases). The innermost circle represents ICD-10 codes with character length 2 and the outmost circle codes with character length 5 (inner circle=length 2, outer circle=length 5) and grouped predictors. Each predictor variable (occurrence, frequency, recency, duration) derived for each ICD-10 code corresponds to a color-coded circle segment. A segment is colored if any coefficient within that group of predictors had a non-zero coefficient for that code, and grey if non-informative. Unspec. = unspecified subcategory.”

Third, with at least 16 supplemental materials, I find this incredibly comprehensive but excessive. Some things can be combined or removed. For example, table S6 and S7 can be combined. Tables S1 and S4 could be removed in my view or made available upon writing to the authors.

Answer: We thank the reviewer for the suggestion and agree. We have removed Tables S1, S3 and S4 from the supplementary materials and combined tables S6 and S7. All removed tables are of course available upon request.

Finally, figure S4 would benefit from a blow-out of the predicted survival in the 0.8-1.0 or 0.9-1.0 range, where 'most of the action is'. This could either be done using a non-linear graph or a graph within graph, at least for the 3 development cohort figures (otherwise we have 9 graphs to revise).

Answer: We agree that the figure can be improved. We had actually revised the figure before but accidentally included the wrong version in the submission. The revised figure is based the same computations that underly Figure 2, and the points are now better spread out (see below).

Reviewer #2: The authors have made improvements to the figures/tables to improve the interpretability of their work. They provide reasonable explanation for the scope and aim of their work in response to some methodological critiques, especially in light of their ongoing follow-up studies within this domain.

Answer: We thank the reviewer for this feedback. 

In conclusion we thank the Editor and the Reviewers for the suggestions that have led to further improvements of the manuscript and hope that it is now suitable to be considered for publication in PLOS ONE.

Marcus Westerberg, PhD

Department of Surgical Sciences, 

Uppsala University

e-mail: marcus.westerberg@uu.se

---

## [Decision Letter · Decision Letter 2]

19 Dec 2023

Development and validation of a multi-dimensional diagnosis-based comorbidity index that improves prediction of death in men with prostate cancer: Nationwide, population-based register study.

PONE-D-23-15816R2

Dear Dr. Westerberg,

We’re pleased to inform you that your manuscript has been judged scientifically suitable for publication and will be formally accepted for publication once it meets all outstanding technical requirements.

Kind regards,

Eugenio Paci, MD

Academic Editor

PLOS ONE

Additional Editor Comments (optional):

This is a very well documented and relevant paper. The process of revision has been very careful and informative and I congratulate with the authors for their contribution in this area, extremely important for the evaluation of screening and clinical interventions. As the authors acknowledged in the discussion section, the national information system in Sweden is the top, however the comparison with the Charlson Index will be a great contribution in understanding the limitations of the measures in many countries  are  currently used and helpful to progress toward more performant systems. 

Reviewers' comments:

Reviewer's Responses to Questions

**Comments to the Author**

1. If the authors have adequately addressed your comments raised in a previous round of review and you feel that this manuscript is now acceptable for publication, you may indicate that here to bypass the “Comments to the Author” section, enter your conflict of interest statement in the “Confidential to Editor” section, and submit your "Accept" recommendation.

Reviewer #1: All comments have been addressed

2. Is the manuscript technically sound, and do the data support the conclusions?

Reviewer #1: Yes

3. Has the statistical analysis been performed appropriately and rigorously? 

Reviewer #1: Yes

4. Have the authors made all data underlying the findings in their manuscript fully available?

Reviewer #1: Yes

5. Is the manuscript presented in an intelligible fashion and written in standard English?

Reviewer #1: Yes

6. Review Comments to the Author

Reviewer #1: The authors have done a very nice job responding to my remaining comments. I appreciate their detailed explanations and attention to details.

7. PLOS authors have the option to publish the peer review history of their article (what does this mean?). If published, this will include your full peer review and any attached files.

Reviewer #1: No

---

## [Editor Report · Acceptance letter]

9 Jan 2024

PONE-D-23-15816R2 

PLOS ONE

Dear Dr. Westerberg, 

I'm pleased to inform you that your manuscript has been deemed suitable for publication in PLOS ONE. Congratulations! Your manuscript is now being handed over to our production team.

Kind regards, 

on behalf of

Dr. Eugenio Paci 

Academic Editor

PLOS ONE